# Sublinear Time Quantum Algorithm for Attention Approximation

**Zhao Song**
Simons Institute for the Theory of Computing, UC Berkeley
magic.linuxkde@gmail.com

**Jianfei Xue**
New York University
jx898@nyu.edu

**Jiahao Zhang**
ml.jiahaozhang02@gmail.com

**Lichen Zhang**
MIT CSAIL
lichenz@csail.mit.edu

## Abstract

Given the query, key and value matrices $Q, K, V \in \mathbb{R}^{n \times d}$, the attention module is defined as $\mathrm{Att}(Q, K, V) = D^{-1}AV$ where $A = \exp(QK^\top / \sqrt{d})$ with $\exp(\cdot)$ applied entrywise, $D = \mathrm{diag}(A\mathbf{1}_n)$. The attention module is the backbone of modern transformers and large language models, but explicitly forming the softmax matrix $D^{-1}A$ incurs $\Omega(n^2)$ time, motivating numerous approximation schemes that reduce runtime to $\widetilde{O}(nd)$ via sparsity or low-rank factorization.

We propose a quantum data structure that approximates any row of $\mathrm{Att}(Q, K, V)$ using only row queries to $Q, K, V$. Our algorithm preprocesses these matrices in $\widetilde{O}\left(\epsilon^{-1} n^{0.5} \left(s_\lambda^{2.5} + s_\lambda^{1.5}d + \alpha^{0.5}d\right)\right)$ time, where $\epsilon$ is the target accuracy, $s_\lambda$ is the $\lambda$-statistical dimension of the exponential kernel defined by $Q$ and $K$, and $\alpha$ measures the row distortion of $V$ that is at most $d/\mathrm{srank}(V)$, the stable rank of $V$. Each row query can be answered in $\widetilde{O}(s_\lambda^2 + s_\lambda d)$ time.

To our knowledge, this is the first quantum data structure that approximates rows of the attention matrix in sublinear time with respect to $n$. Our approach relies on a quantum Nyström approximation of the exponential kernel, quantum multivariate mean estimation for computing $D$, and quantum leverage score sampling for the multiplication with $V$.

## 1 Introduction

Transformers (Vaswani et al., 2017) have emerged as one of the most successful machine learning architectures in recent years, revolutionizing fields such as natural language processing (Devlin et al., 2019; Yang et al., 2019; Raffel et al., 2020; Brown et al., 2020; Jiao et al., 2020), computer vision (Carion et al., 2020; Dosovitskiy et al., 2021; Guo et al., 2022), speech recognition (Chorowski et al., 2015; Wang et al., 2021), robotics (Liu et al., 2022), and time series forecasting (Zhou et al., 2021). These models typically operate on sequences of length $n$, autoregressively predicting the next most likely token to produce an output of length $n$. In applications like large language models (LLMs), it has been widely observed that increasing the sequence length $n$ significantly enhances generative performance. However, this benefit comes at a substantial computational cost: the core attention module has a quadratic time complexity in $n$, which severely limits both training and inference scalability.

Formally, let $Q, K, V \in \mathbb{R}^{n \times d}$ denote the query, key, and value embeddings. The attention module is defined as

$$\mathrm{Att}(Q, K, V) = D^{-1}AV \in \mathbb{R}^{n \times d},$$

where $A = \exp(QK^\top / \sqrt{d}) \in \mathbb{R}^{n \times n}$ is computed entrywise, and $D = \mathrm{diag}(A\mathbf{1}_n) \in \mathbb{R}^{n \times n}$. The matrix $A$ is referred to as the *attention matrix*, and $D^{-1}A$ as the *softmax matrix*. Due to the $n \times n$ size of $A$, much recent research has focused on reducing the quadratic complexity by approximating attention through pattern-based sparse attention (Daras et al., 2020; Kitaev et al., 2020; Roy et al.,

2021; Sun et al., 2022; Child et al., 2019; Beltagy et al., 2020; Ainslie et al., 2020; Zaheer et al., 2020), linearizing the kernel through feature mapping (Katharopoulos et al., 2020; Choromanski et al., 2021; Wang et al., 2020; Peng et al., 2021), or various algorithmic and data structure optimizations (Zandieh et al., 2023; Alman & Song, 2023; Han et al., 2024; Kacham et al., 2024; Zandieh et al., 2024; van den Brand et al., 2024; Song et al., 2024; Kannan et al., 2025; Chu et al., 2024; Chen et al., 2025b; Indyk et al., 2025).

The theoretical goal in these efforts is to achieve a runtime that scales nearly linearly with $n$, allowing some approximation error. This is a natural target, since the input size to the attention module is $n \times d$. On a classical computer, any algorithm that approximates attention in time $\widetilde{O}(nd)$ is considered optimal. But could this process be accelerated further using a quantum computer?

If our objective is to output the entire $n \times d$ matrix $\mathrm{Att}(Q, K, V)$, then $\Omega(nd)$ time is unavoidable due to output size. However, in many transformer applications — particularly during inference (Pope et al., 2023; Brandon et al., 2024; Adnan et al., 2024; Zhang et al., 2024a; Feng et al., 2025; Liu et al., 2024b; Kumari et al., 2024; Behnam et al., 2025; Chen et al., 2025a;c; Indyk et al., 2025) — only *row queries* are needed. In this setting, we aim to preprocess $Q, K, V$ into a data structure such that, for any index $i \in [n]$, the structure can return a vector $\widetilde{r}_i \in \mathbb{R}^d$ that approximates the $i$-th row of $\mathrm{Att}(Q, K, V)$. This model circumvents the $\Omega(nd)$ lower bound by focusing on partial output. Nonetheless, since each row of $\mathrm{Att}(Q, K, V)$ is a convex combination of rows of $V$, achieving truly sublinear time in $n$ still appears classically intractable.

In this work, we answer this question affirmatively. Specifically, we construct a quantum data structure that preprocesses $Q, K, V$ using only *row queries*, and does so in time[1] $\widetilde{O}(\epsilon^{-1} n^{0.5} \cdot \mathrm{poly}(d, s_\lambda, \alpha))$, where $s_\lambda$ is the *statistical dimension* of the exponential kernel matrix associated with $Q$ and $K$, and $\alpha$ is a measure of the row distortion of $V$ (see Definition C.2). Given any index $i \in [n]$, the data structure returns an approximation to the $i$-th row of $\mathrm{Att}(Q, K, V)$ in time $\widetilde{O}(s_\lambda^2 + s_\lambda d)$.

To our knowledge, this is the first quantum algorithm to implement the row query model in sublinear time. Prior works either require superlinear preprocessing time or impose structural assumptions (Gao et al., 2023). Our approach avoids both: it makes *no assumptions* on $Q, K, V$, making it broadly applicable in practice. Moreover, our construction is conceptually simple — it combines quantum techniques such as Grover search (Grover, 1996), Nyström kernel approximation, and quantum multivariate mean estimation (Cornelissen et al., 2022) to approximate each component of the attention module: $D$, $A$, and $V$.

**Quantum Computation Model.** We follow the standard quantum computation framework as in Apers & De Wolf (2022); Apers & Gribling (2023). The model allows quantum subroutines using $O(\log n)$ qubits, quantum queries to the input, and access to a quantum-read/classical-write RAM (QRAM) of $\mathrm{poly}(n)$ bits. Each quantum read or classical write takes unit cost. We measure *time complexity* by the number of QRAM operations, and *query complexity* by the number of queries to the input. In our setting, we query rows of $Q$, $K$, and $V$, each requiring $O(d)$ time classically. For simplicity, we assume $Q$ and $K$ have been scaled by $1/d^{1/4}$, which can also be done via row queries in $O(d)$ time.

## 2 PRELIMINARY

**Notation.** Given symmetric matrices $A, B \in \mathbb{R}^{n \times n}$, we use $A - B \succeq 0$ to denote $A - B$ is a positive semidefinite (PSD) matrix, i.e., for any $x \in \mathbb{R}^n$, $x^\top (A - B) x \geq 0$. Given a matrix $M \in \mathbb{R}^{n \times n}$, we use $\exp(M)$ to denote the entrywise exponentiation operation. We use $\mathrm{tr}[M]$ to denote the trace of $M$. For a real matrix $A$, we use $A^\dagger$ to denote its Moore-Penrose pseudoinverse, and for a square, nonsingular real matrix $M$, we use $M^{-1}$ to denote its inverse. For two vectors $x, y \in \mathbb{R}^n$, we use $x^\top y$ or $\langle x, y \rangle$ to denote the inner product of $x$ and $y$. We use $\mathbf{0}_n$ and $\mathbf{1}_n$ to denote all-0's and all-1's vector. For a vector $x \in \mathbb{R}^n$, we use $\|x\|_2 = \sqrt{x^\top x}$ to denote its $\ell_2$ norm, $\|x\|_\infty = \max_{i \in [n]} |x_i|$ to denote its $\ell_\infty$ norm. If $M$ is a PSD matrix, then we use $\|x\|_M = \sqrt{x^\top M x}$ to denote the $M$-energy norm of $x$. For a matrix $A$, we use $\|A\|$ to denote its spectral norm and $\|A\|_\infty$ to denote its max row $\ell_1$ norm, and $\|A\|_F$ to denote its Frobenius norm. Throughout the paper, we will also exclusively

---

[1] We use $\widetilde{O}(\cdot)$ to suppress polylogarithmic factors in $n$, $d$, $s_\lambda$, and $1/\epsilon$.

work with weighted sampling matrices, usually denoted by $S \in \mathbb{R}^{n \times s}$ for where $s$ is the total number of samples taken, let $i(j)$ be the index of the $i$-th sample, then the $i$-th column of $S$ is $\frac{1}{\sqrt{p_j}} e_j$, where $p_j$ is the probability of choosing the index $j$. We use $\mathbb{E}[X]$ to denote the expectation of a random variable $X$. We use $\mathbb{I}[E]$ to denote the indicator of whether event $E$ happens.

**Numerical Linear Algebra.** We rely on several primitives from numerical linear algebra for fast approximations and provable guarantees.

**Definition 2.1** (Leverage score). *Let $A \in \mathbb{R}^{n \times d}$. The $i$-th leverage score of $A$ is defined as*

$$\tau_i := a_i^\top (A^\top A)^{-1} a_i,$$

*where $a_i$ is the $i$-th row of $A$. Equivalently, let $A = U\Sigma V^\top$ be its SVD, then $\tau_i = \|u_i\|_2^2$, where $u_i$ is the $i$-th row of $U$.*

We will also work exclusively with *kernel matrices*. Given a dataset $X = \{x_1, \ldots, x_n\} \subseteq \mathbb{R}^d$, we define the exponential kernel matrix $E \in \mathbb{R}^{n \times n}$ by $E_{i,j} = \exp(\langle x_i, x_j \rangle)$. Although $E$ is generally full-rank, our algorithm depends only on a parameter called the $\lambda$-*statistical dimension* of $E$, which may be much smaller than $n$.

**Definition 2.2** (Statistical dimension (Zhang, 2005; Hastie et al., 2009)). *Let $E \in \mathbb{R}^{n \times n}$ be a PSD matrix, and let $\lambda > 0$. The $\lambda$-statistical dimension of $E$ is defined as $s_\lambda(E) := \mathrm{tr}[E(E + \lambda I)^{-1}]$. When $E$ is clear from context, we write $s_\lambda$ for simplicity.*

Note that $s_\lambda$ is a monotonically decreasing function of $\lambda$, and is closely related to the notion of ridge leverage scores.

**Definition 2.3** (Ridge leverage score (Alaoui & Mahoney, 2015)). *Let $E \in \mathbb{R}^{n \times n}$ be a kernel matrix and let $\lambda > 0$. The $\lambda$-ridge leverage score of the data point $x_i$ is defined as*

$$\tau_i^\lambda := (E(E + \lambda I)^{-1})_{i,i}.$$

*If $E = BB^\top$ for some $B \in \mathbb{R}^{n \times n}$, then this can be equivalently written as*

$$\tau_i^\lambda = b_i^\top (B^\top B + \lambda I)^{-1} b_i,$$

*where $b_i$ is the $i$-th row of $B$.*

It is easy to see that $\sum_{i=1}^n \tau_i^\lambda = s_\lambda$. Moreover, Musco & Musco (2017) shows that Nyström approximations (Williams & Seeger, 2000) based on ridge leverage score sampling yield accurate spectral approximations to $E$.

**Lemma 2.4** (Theorem 3 of Musco & Musco (2017)). *Let $s = O(s_\lambda \log(s_\lambda/\delta))$, $\lambda > 0$, and $\delta \in (0, 1)$. Let $E \in \mathbb{R}^{n \times n}$ be any kernel matrix. Let $S \in \mathbb{R}^{n \times s}$ be the $\lambda$-ridge leverage score sampling matrix. Then the Nyström approximation $\widetilde{E} := ES(S^\top ES)^\dagger S^\top E$ satisfies $E \preceq \widetilde{E} \preceq E + \lambda I$ with probability at least $1 - \delta$.*

**Quantum Primitives.** In this paper, we primarily leverage two quantum algorithmic primitives. The first is an efficient quantum sampling oracle based on Grover search.

**Lemma 2.5** (Claim 3 in Apers & De Wolf (2022)). *Let $n$ be a positive integer, and let $\{p_1, \ldots, p_n\} \subseteq [0, 1]$ be a list of probabilities. There exists a quantum algorithm, $\mathrm{QSAMPLE}(p)$, that generates a list of indices where each $i$ is sampled independently with probability $p_i$, in time $\widetilde{O}\left(\sqrt{n \sum_{i=1}^n p_i}\right) \cdot \mathcal{T}$, where $\mathcal{T}$ denotes the time required to generate any individual $p_i$.*

The second primitive is a quantum procedure for approximating matrix-vector products using quantum multivariate mean estimation.

**Lemma 2.6** (Theorem 5.1 of Apers & Gribling (2023)). *Let $\epsilon \in (0, 1)$, and let $A \in \mathbb{R}^{n \times d}$ and $v \in \mathbb{R}^n$. Suppose we are given quantum query access to the rows of $A$ and the entries of $v$. Then there exists a quantum algorithm $\mathrm{QMATVEC}(A, v, \epsilon)$ that outputs a vector $\widetilde{\mu} \in \mathbb{R}^d$ such that, with probability at least $1 - 1/\mathrm{poly}(n)$, $\|\widetilde{\mu} - A^\top v\|_{(A^\top A)^{-1}} \leq \epsilon$, using $\widetilde{O}\left(\epsilon^{-1} n^{0.5} d^{0.5} \|v\|_\infty\right)$ queries to $A$ and $v$.*

## 3 TECHNICAL OVERVIEW

In this section, we provide an overview on the algorithmic techniques we utilize to approximate $A, D$ and $V$, in sublinear time.

### 3.1 APPROXIMATE THE ATTENTION MATRIX VIA QUANTUM NYSTRÖM

To approximate the attention matrix $A$, we will make use of Nyström approximation (Williams & Seeger, 2000). However, recall that $A = \exp(QK^\top)$; for $Q \neq K$, the matrix itself is not even symmetric. This poses significant challenges for obtaining a good approximation. On the other hand, if we treat the queries and keys as the *dataset*, and form the exponential kernel matrix over them, then the resulting matrix is indeed a kernel matrix.

Specifically, let the dataset $X = \{q_1, \ldots, q_n, k_1, \ldots, k_n\}$, and consider $E \in \mathbb{R}^{2n \times 2n}$ where $E = \begin{bmatrix} \exp(QQ^\top) & \exp(QK^\top) \\ \exp(KQ^\top) & \exp(KK^\top) \end{bmatrix}$, then the attention matrix can be retrieved via $PE \begin{bmatrix} \mathbf{0}_n \\ \mathbf{1}_n \end{bmatrix}$ where $P \in \mathbb{R}^{n \times 2n}$ is the matrix consisting of the first $n$ rows of the $2n \times 2n$ identity matrix, which selects the first $n$ rows of $E$. Thus, once we obtain an approximation for $E$, we automatically obtain an approximation for $A$.

It remains to compute a Nyström approximation of $E$, as at first glance it is not clear how to even generate the ridge leverage score sampling matrix $S$ in sublinear time. Musco & Musco (2017) shows that on a classical computer, it is possible to compute a *generalized* ridge leverage score sampling matrix using $\widetilde{O}(ns_\lambda)$ evaluations of the kernel function and an additional $\widetilde{O}(ns_\lambda^2)$ time, via a recursive sampling scheme:

- Uniformly sample half of the data points, then recursively compute the weighted sampling matrix $\widetilde{S}^{n \times s}$ for the subset;

- Compute the *generalized ridge leverage score*, defined as $\widetilde{\tau}_i^\lambda := b_i^\top (B^\top \widetilde{S}\widetilde{S}^\top B + \lambda I)^\dagger b_i$, and set $p_i = \min\{1, \widetilde{\tau}_i^\lambda \cdot \log(s_\lambda/\delta)\}$;

- Output $S$ as the weighted sampling matrix according to $p_i$.

The key ingredients in their algorithm are (1) the generalized ridge leverage score can be computed via kernel function evaluations instead of computing the factorization (see Definition A.4), and (2) sampling according to generalized ridge leverage score only increases the sample size by a constant factor, hence it does not affect the asymptotic runtime of the algorithm (see Lemma A.3).

For the simpler setting of leverage score sampling, Apers & Gribling (2023) shows that this recursive framework can benefit from quantum speedup, especially the Grover search sampler of Lemma 2.5, by noting that when sampling according to the leverage score, it is not necessary to compute or approximate all the scores; rather, it is enough to implement an oracle that can supply any approximate leverage score when needed.

For our application, however, this oracle is much more difficult to implement, as in the setting of Apers & Gribling (2023), one could directly query the row of $B$, which is not the case for the kernel setting. Nevertheless, we show how to implement such an oracle for generalized ridge leverage scores of kernels. The algorithm is detailed in Algorithm 1. Throughout this section, we let $s$ denote the final sample size of the Nyström approximation.

The main idea is to utilize the identity $\widetilde{\tau}_i^\lambda = \frac{1}{\lambda}(E - ES(S^\top ES + \lambda I)^{-1}S^\top E)_{i,i}$, where $E_{i,i}$ involves a single kernel evaluation $\mathsf{K}(x_i, x_i)$, and $S^\top ES$ requires only $O(s^2)$ kernel evaluations. Finally, the term $(ES(S^\top ES + \lambda I)^\dagger S^\top E)_{i,i}$ can be computed by evaluating the kernel between $x_i$ and the sampled points in $S$, weighted appropriately, which requires $O(s)$ kernel evaluations. This shows that we can implement the oracle by precomputing $(S^\top ES + \lambda I)^\dagger$ in $O(s^2) \cdot \mathcal{T}_\mathsf{K} + s^\omega$ time, where $\mathcal{T}_\mathsf{K}$ denotes the time for kernel evaluation and $\omega \approx 2.37$ is the matrix multiplication exponent (Duan et al., 2023; Williams et al., 2024; Alman et al., 2025). Each oracle query can then be answered in $O(s) \cdot \mathcal{T}_\mathsf{K} + s^2$ time. By Lemma 2.5, the quantum sampler requires only $\widetilde{O}(n^{0.5}s^{0.5})$ oracle calls, so the overall runtime is $\widetilde{O}(n^{0.5}s^{1.5} \cdot (\mathcal{T}_\mathsf{K} + s) + s^\omega)$. In our setting, the

---

**Algorithm 1** Quantum Nyström approximation via recursive generalized ridge leverage score sampling.

---

1: **procedure** QNYSTRÖMKERNEL($\{x_1, \ldots, x_n\} \in (\mathbb{R}^d)^n, \mathsf{K} : \mathbb{R}^d \times \mathbb{R}^d \to \mathbb{R}^m, \delta \in (0,1), \lambda \in (0, \infty)$)       $\triangleright$ $\delta$ is the failure probability, $\lambda$ is the ridge leverage score parameter.
2:      $s \leftarrow O(s_\lambda \log(s_\lambda/\delta))$
3:      $T \leftarrow O(\log(n/s))$
4:      Let $S_0 \subset_{1/2} S_1 \subset_{1/2} \cdots \subset_{1/2} S_T = [n]$      $\triangleright$ We use $A \subset_{1/2} B$ to denote $A$ is a uniform subset of half of the indices of $B$
5:      $M_0 \leftarrow \{\mathsf{K}(x_i, x_j)\}_{(i,j) \in S_0 \times S_0}$      $\triangleright$ $|S_0| = s$
6:      Let $D_0 \in \mathbb{R}^{n \times |S_0|}$ be the sampling matrix of $S_0$
7:      **for** $t = 1$ to $T$ **do**
8:          $\widehat{M} \leftarrow (M_{t-1} + \lambda I_s)^{-1}$
9:          $\triangleright$ Let $D_{t-1}^\top K_i := \{D_{t-1}(j) \cdot \mathsf{K}(x_i, x_j)\}_{j \in D_{t-1}} \in \mathbb{R}^s$ for $i \in S_t$ where $D_{t-1}(j)$ is the weight corresponding to $x_j$ specified by $D_{t-1}$
10:          Implement oracle for $q_i \leftarrow \frac{5}{\lambda} \cdot (\mathsf{K}(x_i, x_i) - (D_{t-1}^\top K_i)^\top \widehat{M} D_{t-1}^\top K_i)$ for $i \in S_t$
11:          $\triangleright$ $p_i = \min\{1, 16 q_i \log(2s/\delta)\}$
12:          $\widetilde{D}_t \leftarrow$ QSAMPLE($p$)      $\triangleright$ $\widetilde{D}_t \in \mathbb{R}^{|S_t| \times s}$
13:          $D_t \leftarrow D_{S_t} \cdot \widetilde{D}_t$      $\triangleright$ $D_t \in \mathbb{R}^{n \times s}$
14:          $M_t \leftarrow \{\widetilde{D}_t(i) D_t(j) \cdot \mathsf{K}(x_i, x_j)\}_{(i,j) \in D_t \times D_t}$      $\triangleright$ $M_t \in \mathbb{R}^{s \times s}$
15:      **end for**
16:      **return** $D_T$
17: **end procedure**

---

kernel function $\mathsf{K}(x_i, x_j) = \exp(\langle x_i, x_j \rangle)$ can be computed in $O(d)$ time, which gives a runtime of $\widetilde{O}(n^{0.5} s^{1.5}(d + s) + s^\omega)$, sublinear in $n$.

It remains to analyze the approximation guarantee. Sampling according to generalized ridge leverage scores ensures that $E \preceq \widetilde{E} \preceq E + \lambda I$, but this does not immediately imply a bound on the approximation error for $\exp(QK^\top)$. To address this, let $E = \begin{bmatrix} B & A \\ A^\top & C \end{bmatrix}$ and $\widetilde{E} = \begin{bmatrix} \widetilde{B} & \widetilde{A} \\ \widetilde{A}^\top & \widetilde{C} \end{bmatrix}$.

Standard spectral approximation theory guarantees that $B \preceq \widetilde{B} \preceq B + \lambda I$ and $C \preceq \widetilde{C} \preceq C + \lambda I$. For the off-diagonal block we are interested in $A$, we cannot get such a strong spectral approximation guarantee; in fact, one can show that the best we could hope for is a symmetrization bound: $A + A^\top \preceq \widetilde{A} + \widetilde{A}^\top \preceq A + A^\top + 2\lambda I$. On the other hand, a weaker and a more handy bound can be exhibited: $\|A - \widetilde{A}\| \leq \lambda$ and $\|A - \widetilde{A}\|_F \leq \lambda \sqrt{n}$, and we will show these bounds are sufficient to derive the final approximation guarantees of our algorithm.

It is also worth noting that Algorithm 1 merely computes the weighted sampling matrix $S$, which can be stored compactly by recording the sampled indices and corresponding weights, but does not explicitly form the Nyström approximation $\widetilde{E} = ES(S^\top ES)^\dagger S^\top E$. While $(S^\top ES)^\dagger$ can be computed and stored in $O(s^2 d + s^\omega)$ time, forming $\widetilde{E}$ would take $\Omega(ns)$ time, which is prohibitive due to output size. In what follows, we show that this restricted representation of $S$ is nonetheless sufficient to approximate $D$, $V$, and $\mathrm{Att}(Q, K, V)$.

We now compare our Nyström approximation scheme to a related method known as Nyström-former (Xiong et al., 2021), which also integrates Nyström into the attention mechanism. Specifically, they consider the attention matrix $A$ and partition it as $A = \begin{bmatrix} X_1 & X_2 \\ X_3 & X_4 \end{bmatrix}$, aiming to approximate $X_4$ using the other three blocks. Given Nyström landmark points $Q'$ and $K'$ sampled from $Q$ and $K$, they set $X_1 = \exp(Q'K'^\top)$, $X_2 = \exp(QK'^\top)$, and $X_3 = \exp(Q'K^\top)$. Since the number of landmarks is small, these blocks are all low-dimensional. Xiong et al. (2021) proves that $X_4$ can be efficiently approximated using $X_1$, $X_2$, and $X_3$ in $O(nmd)$ time, where $m$ is the number of landmarks. While Nyströmformer performs well in practice, it guarantees convergence to the true attention matrix only when all rows of $Q$ and $K$ are included as landmarks. In contrast, our Nyström scheme operates on

the exponential kernel matrix formed from $Q$ and $K$, and achieves spectral approximation guarantees as long as the sample size is sufficiently large without needing to include all data points.

## 3.2 Approximate the Normalization Factor via Quantum Mean Estimation

Recall that $D = \mathrm{diag}(A\mathbf{1}_n)$, and each normalization factor only requires computing $a_i^\top \mathbf{1}_n$, where $a_i$ is the $i$-th row of $A$. If we have access to $\widetilde{E}$, then the $i$-th normalization factor could be estimated as $\widetilde{E}_{i,*}^\top \begin{bmatrix} \mathbf{0}_n \\ \mathbf{1}_n \end{bmatrix}$. However, as discussed earlier, we cannot explicitly form $\widetilde{E}$ due to its size. To resolve this, we define $U := ES(S^\top ES)^{\dagger/2} \in \mathbb{R}^{2n \times s}$. By the definition of the Nyström approximation, we have $\widetilde{E} = UU^\top$. Moreover, $U$ also exhibits a block structure $U = \begin{bmatrix} U_1 \\ U_2 \end{bmatrix}$ where $U_1, U_2 \in \mathbb{R}^{n \times s}$, and the desired approximate $\widetilde{A} = U_1 U_2^\top$ can be obtained via these blocks. Given any vector $v \in \mathbb{R}^n$, if we can compute or approximate $U_2^\top v$, then the normalization factor for the $i$-th row can be estimated as $(U_1)_{i,*}^\top (U_2^\top v)$ where $(U_1)_{i,*} \in \mathbb{R}^s$ is the $i$-th row of $U_1$. Fortunately, we can implement row queries to $U_2$. We first precompute $(S^\top ES)^{\dagger/2}$ in $O(s^2 d + s^\omega)$ time, then each row $(U_2)_{i,*}$ of $U_2$ is computed via kernel evaluations between $x_{i+n}$ and the points in $S$, followed by matrix-vector multiplication with $(S^\top ES)^{\dagger/2}$. This takes $O(s^2 + sd)$ time.

It remains to approximate $U_2^\top v$, which we cast as a multivariate mean estimation problem. Define the random variable $X = 2n v_i (U_2)_{*,i}$, where $i \in [n]$ is selected uniformly at random. It is easy to verify that $\mathbb{E}[X] = U_2^\top v$, and the variance is bounded. Therefore, one can apply the quantum multivariate mean estimation procedure of Cornelissen et al. (2022) to approximate $U_2^\top v$. To further reduce variance, Apers & Gribling (2023) proposes approximating the matrix-vector product in the $(U_2^\top U_2)^{-1}$-energy norm. Following this idea, we apply Lemma 2.6 to output a vector $\widetilde{\mu} \in \mathbb{R}^s$ such that $\|\widetilde{\mu} - U_2^\top v\|_{(U_2^\top U_2)^{-1}} \le \epsilon$, using $\widetilde{O}(\epsilon^{-1} n^{0.5} s^{0.5} \|v\|_\infty)$ row queries to $U_2$ and $v$. In our application, we always have $\|v\|_\infty = 1$, and as noted above, each row query to $U$ takes $O(s^2 + sd)$ time. We present the full algorithm below in Algorithm 2.

For the approximation guarantee, we prove that for any vector $x \in \mathbb{R}^s$, if $\|x\|_{(U_2^\top U_2)^{-1}} \le \epsilon$, then $\|U_1 x\|_2 \le \epsilon \cdot \|U_1 U_2^\top\|$. This is particularly useful for us, as we can set $x = U_2^\top v - \widetilde{\mu}$, in which case $U_1 x = U_1 U_2^\top v - U_1 \widetilde{\mu} = \widetilde{A}v - U_1 \widetilde{\mu}$, and the upper bound becomes $\epsilon \cdot \|\widetilde{A}\| \le \epsilon \cdot (\|A\| + \lambda)$. On the other hand, we can upper bound $\|(\widetilde{A} - A)v\|_\infty$ using the matrix infinity norm, defined as $\|\widetilde{A} - A\|_\infty = \max_{i \in [n]} \|\widetilde{A}_{i,*} - A_{i,*}\|_1$. A simple argument shows that $\|\widetilde{A} - A\|_\infty \le \sqrt{n} \cdot \|\widetilde{A} - A\| \le \lambda\sqrt{n}$. A triangle inequality then yields the final approximation guarantee. If we define $\widetilde{D} := \mathrm{diag}(\widetilde{A}\mathbf{1}_n)$, the above analysis provides a bound on $\|D - \widetilde{D}\|$. However, in forming the attention module, it is more desirable to control $\|\widetilde{D}^{-1}\|$. To achieve this, we prove a perturbation bound on matrix inversion that relates $\|\widetilde{D}^{-1}\|$ to $\|D^{-1}\|$.

## 3.3 Approximate the Value Matrix via Leverage Score Sampling

In preceding discussions, we have shown how to construct the sampling matrix for Nyström approximation and how to compute the normalization factor for any row $i \in [n]$. It remains to approximate $V$ in sublinear time. Prior classical algorithms, such as Zandieh et al. (2023), propose using importance sampling based on the *joint row norm* of $V$ and $D^{-1}A$. Specifically, the sampling probability for the $i$-th row is set as $p_i \ge 1/4 \cdot (\|e_i^\top D^{-1}A\|_2^2 + \gamma \cdot \|v_i\|_2^2)/(\|D^{-1}A\|_F^2 + \gamma \cdot \|V\|_F^2)$, where $\gamma = \|D^{-1}A\|^2 / \|V\|^2$. This method achieves a final sample size that is nearly linear in $d + \mathrm{srank}(D^{-1}A)$, where $\mathrm{srank}(D^{-1}A) = \|D^{-1}A\|_F^2 / \|D^{-1}A\|^2$ is the stable rank of the softmax matrix. While this approach is conceptually simple and easy to implement, it requires estimating the Frobenius norms of both $V$ and $D^{-1}A$ to constant-factor accuracy. This is straightforward if we are allowed to read all entries of $V$, but becomes particularly challenging in sublinear time. Our solution is to instead use leverage score sampling on the matrix $V$, which can be implemented in sublinear time (Apers & Gribling, 2023).

Unlike the joint sampling distribution of Zandieh et al. (2023), which yields a *spectral norm approximate matrix multiplication* guarantee of the form $\|D^{-1}ASS^\top V\| \le \epsilon \cdot \|D^{-1}A\| \cdot \|V\|$, leverage

---

**Algorithm 2** Algorithm for estimating normalization factor.

1: **data structure** QRowNorm
2: **begin members**
3:     $s \in \mathbb{N}$
4:     $S \in (\mathbb{R}^2)^s$
5:     $N \in \mathbb{R}^{s \times s}$
6:     $\widetilde{\mu} \in \mathbb{R}^s$
7: **end members**
8:
9: **procedure** PREPROCESS($Q \in \mathbb{R}^{n \times d}, K \in \mathbb{R}^{n \times d}, \lambda \in (0, \infty), \epsilon \in (0, 1)$)
10:     $s \leftarrow O(s_\lambda \log(s_\lambda n))$
11:     $S \leftarrow$ QNyströmKernel($Q \cup K, (x_i, x_j) \mapsto \exp(\langle x_i, x_j \rangle), 1/\operatorname{poly}(n), \lambda$)        ▷
    Algorithm 1, $S$ is a list of sampled indices and weights
12:     $N \leftarrow (S^\top E S)^{\dagger/2}$
13:     Implement row oracle $(U_2)_{j,*}$ as follows:
14:         $\widetilde{(U_2)}_{j(k),*} \leftarrow S_k \cdot \exp(\langle x_{j+n}, x_k \rangle), \forall k \in S$                     ▷ $\widetilde{(U_2)}_{j(k),*} \in \mathbb{R}^s$
15:         $(U_2)_{j,*} \leftarrow N \widetilde{(U_2)}_{j(k),*}$        ▷ $S$ stores pairs of indices and weights, $S_k$ is the weight
    corresponding to index $k$, $(U_2)_{j,*} \in \mathbb{R}^s$
16:     Implement entry oracle for a vector $v = \mathbf{1}_n \in \mathbb{R}^n$
17:     $\widetilde{\mu} \leftarrow$ QMatVec($U_2, v, \epsilon$)                          ▷ $\widetilde{\mu} \in \mathbb{R}^s$, Lemma 2.6
18: **end procedure**
19:
20: **procedure** QUERY($i \in [n]$)
21:     $b_i \leftarrow \langle (U_1)_{i,*}, \widetilde{\mu} \rangle$                          ▷ $(U_1)_{i,*}$ is computed via row oracle
22:     **return** $b_i$
23: **end procedure**
24: **end data structure**

---

score sampling has two key limitations: (1) it requires that $V$ have orthonormal columns (Clarkson & Woodruff, 2017), and (2) it provides approximate matrix multiplication guarantees in Frobenius norm, i.e., $\|D^{-1}ASS^\top V\|_F \leq \epsilon \cdot \|D^{-1}A\|_F \cdot \|V\|_F$.

To address the first limitation, we introduce a new parameter called the *row distortion* of $V$, defined as $\alpha := d/\|V\|_F^2 \cdot \max_{i \in [n]} \|v_i\|_2^2/\tau_i$. Intuitively, $\alpha$ measures the mismatch between the row density and row importance. Specifically, the ratio $\|v_i\|_2^2/\|V\|_F^2$ quantifies how much row $v_i$ contributes in $\ell_2^2$ norm, while $\tau_i/d$ measures how linearly independent $v_i$ is compared to other rows via $\tau_i$.

Our main result is that by sampling $\widetilde{O}(\epsilon^{-2}\alpha)$ rows of $V$ according to its leverage score distribution, we obtain an approximate matrix multiplication guarantee in Frobenius norm. Note that $\alpha = 1$ if $V$ has orthonormal columns, which recovers the result of Clarkson & Woodruff (2017). This sampling procedure can be implemented in $\widetilde{O}(\epsilon^{-1}n^{0.5}\alpha^{0.5}d)$ time by making row queries to $V$.

### 3.4 MAIN RESULT

Now that we have described how to approximate each of the matrices $D$, $A$, and $V$, we are in a position to state our main result. We provide an overview of our algorithm below in Algorithm 3.

**Theorem 3.1** (Informal version of Theorem D.2). *Let $Q, K, V \in \mathbb{R}^{n \times d}$ be the query, key and value matrices, let $\epsilon, \lambda > 0$. Let $E \in \mathbb{R}^{2n \times 2n}$ be the exponential kernel matrix on the dataset $Q \cup K$ and $s_\lambda$ be the statistical dimension of $E$ (Definition 2.2) and $\alpha$ be the row distortion of $V$ (Definition C.2). Assume that $\|D^{-1}\| < \frac{1}{\epsilon\|A\| + \lambda\sqrt{n}}$ and let $\beta = \frac{1}{1 - (\epsilon\|A\| + \lambda\sqrt{n})\|D^{-1}\|}$. There exists a quantum data structure that preprocesses $Q, K, V$ through only row queries to these matrices and maintains matrices $\widetilde{D}, \widetilde{A}, \widetilde{V}$ implicitly such that, with probability at least $1 - 1/\operatorname{poly}(n)$,*

$$\|\widetilde{D}^{-1}\widetilde{A}\widetilde{V} - \operatorname{Att}(Q, K, V)\|_F \leq \epsilon \cdot (\beta \cdot \|D^{-1}\|) \cdot (\|A\|_F + \lambda\sqrt{n}) \cdot \|V\|_F.$$

*Moreover, the data structure has the specification*

---

**Algorithm 3** Quantum data structure for attention row query.

---

1: **data structure** QATTENTION $\qquad\qquad\qquad\qquad\qquad\qquad$ ▷ Theorem 3.1
2: **begin members**
3: $\quad s_E, s_V \in \mathbb{N}$
4: $\quad \widetilde{V} \in \mathbb{R}^{s_V \times d}$
5: $\quad \widetilde{N} \in \mathbb{R}^{s_E \times s_V}$
6: $\quad \widetilde{L} \in \mathbb{R}^{s_E \times d}$
7: $\quad$ QROWNORM QRN $\qquad\qquad\qquad\qquad\qquad\qquad\qquad$ ▷ Algorithm 2
8: **end members**
9:
10: **procedure** PREPROCESS($Q \in \mathbb{R}^{n \times d}, K \in \mathbb{R}^{n \times d}, V \in \mathbb{R}^{n \times d}, \lambda > 0, \epsilon > 0, \alpha \geq 1$)
11: $\quad s_\lambda \leftarrow s_\lambda(E)$
12: $\quad s_V \leftarrow \widetilde{O}(\epsilon^{-2}\alpha), s_E \leftarrow \widetilde{O}(s_\lambda)$
13: $\quad$ QRN.PREPROCESS($Q, K, \lambda, \epsilon$) $\qquad\qquad\qquad\qquad\qquad$ ▷ Algorithm 2
14: $\quad S_V \leftarrow$ QLEVERAGESCORE($V, s_V$) $\qquad\qquad$ ▷ $S_V \in \mathbb{R}^{n \times s_V}$, Lemma A.5
15: $\quad \widetilde{V} \leftarrow S_V^\top V$ $\qquad\qquad\qquad\qquad\qquad\qquad\qquad$ ▷ $\widetilde{V} \in \mathbb{R}^{s_V \times d}$
16: $\quad S_E \leftarrow$ QNYSTRÖMKERNEL($Q \cup K, (x_i, x_j) \mapsto \exp(\langle x_i, x_j \rangle), 1/\operatorname{poly}(n), \lambda$)
17: $\qquad\qquad\qquad\qquad\qquad\qquad$ ▷ Let $x_1, \ldots, x_{2n}$ denote the dataset $Q \cup K$
18: $\quad \widetilde{M} \leftarrow \{S_E(i)S_E(j) \cdot \exp(\langle x_i, x_j \rangle)\}_{(i,j) \in S_E \times S_E}$ $\qquad$ ▷ $\widetilde{M} \in \mathbb{R}^{s_E \times s_E}$
19: $\quad \widetilde{R} \leftarrow \{S_E(i)S_V(j) \cdot \exp(\langle x_i, x_j \rangle)\}_{(i,j) \in S_E \times S_V}$ $\quad$ ▷ $\widetilde{R} \in \mathbb{R}^{s_E \times s_V}, \widetilde{R} = S_E^\top E \widetilde{S}_V$
20: $\quad \widetilde{N} \leftarrow \widetilde{M}^\dagger \widetilde{R}$ $\qquad\qquad\qquad\qquad\qquad\qquad$ ▷ $\widetilde{N} \in \mathbb{R}^{s_E \times s_V}$
21: $\quad \widetilde{L} \leftarrow \widetilde{N}\widetilde{V}$ $\qquad\qquad\qquad\qquad\qquad\qquad\qquad$ ▷ $\widetilde{L} \in \mathbb{R}^{s_E \times d}$
22: **end procedure**
23:
24: **procedure** QUERY($i \in [n]$)
25: $\quad b_i \leftarrow$ QRN.QUERY($i$) $\qquad\qquad\qquad\qquad\qquad\qquad$ ▷ Algorithm 2
26: $\quad u_i \leftarrow \{S_E(j) \cdot \exp(\langle x_i, x_j \rangle)\}_{j \in S_E}$ $\qquad\qquad\qquad$ ▷ $u_i \in \mathbb{R}^{s_E}$
27: $\quad$ **return** $\widetilde{L}^\top u_i / b_i$
28: **end procedure**
29: **end data structure**

---

- *It preprocesses $Q, K, V$ in $\widetilde{O}(\epsilon^{-1}n^{0.5}s_\lambda^{0.5})$ row queries to $Q, K$ and $\widetilde{O}(\epsilon^{-1}n^{0.5}\alpha^{0.5})$ row queries to $V$, and $\widetilde{O}(\epsilon^{-1}n^{0.5}(s_\lambda^{2.5} + s_\lambda^{1.5}d + \alpha^{0.5}d))$ time;*

- *For any $i \in [n]$, it returns a vector $\widetilde{r}_i = e_i^\top \widetilde{D}^{-1}\widetilde{A}\widetilde{V}$ in $\widetilde{O}(s_\lambda^2 + s_\lambda d)$ time.*

We pause to make some remarks on Theorem 3.1. The preprocessing time scales with $n^{0.5}$, achieving a quadratic speedup with respect to $n$ over any classical algorithm. Several parameters merit further discussion, in particular the statistical dimension $s_\lambda$ and the approximation factor for $\|D^{-1}\|$, denoted by $\beta$. We summarize their relationships as functions of $\lambda$ in Table 1. The row distortion factor $\alpha$ also affects the runtime, and in Appendix C, we prove that $\alpha \leq \frac{d}{\operatorname{srank}(V)}$ where $\operatorname{srank}(V) = \frac{\|V\|_F^2}{\|V\|^2}$ is the stable rank of $V$. This ensures $\alpha \leq d$ and becomes smaller if the value matrix $V$ has close to $d$ stable rank. We empirically verify that (1) the assumption on $\|D^{-1}\|$ is easy to satisfy with wide range of choices for $\epsilon$, (2) the Frobenius norm of $A$ is only a small constant factor of its spectral norm, (3) the row distortion $\alpha = O(1)$ and (4) the infinity norm of $A$ is only a small constant factor of its spectral norm, implying in practice, the additive $\lambda\sqrt{n}$ term is likely to be $O(\lambda)$. We refer to Appendix E for a more detailed section.

| $\lambda$ | $s_\lambda$ | $\frac{1}{\epsilon\|A\| + \lambda\sqrt{n}}$ | $\beta$ |
|---|---|---|---|
| $\uparrow$ | $\downarrow$ | $\downarrow$ | $\uparrow$ |
| $\downarrow$ | $\uparrow$ | $\uparrow$ | $\downarrow$ |

Table 1: Parameters $s_\lambda$, $\frac{1}{\epsilon\|E\| + \lambda\sqrt{n}}$, and $\beta$ as functions of $\lambda$.

**Bit Complexity of Our Algorithm.** Our discussions and results above are grounded in the assumption that arithmetic operations are performed in infinite precision, while this is usually adopted in the analysis of classical algorithms, QRAM model only allows $O(\log n)$ qubits and $\text{poly}(n)$ bits. In Section F, we provide a preliminary bit complexity analysis of our algorithm, in particular centering around the matrix inversion and pseudoinversion operations. To the best of our knowledge, there is no prior work on studying the bit complexity of numerical linear algebra operations in the QRAM model, and we leave a comprehensive analysis of bit complexity as a future direction.

## 4 RELATED WORK

**Transformers and Attention Mechanism.** Transformers (Vaswani et al., 2017) have been the driving force behind large language models (Devlin et al., 2019; Brown et al., 2020; Touvron et al., 2023; Bubeck et al., 2023; Team et al., 2023; Liu et al., 2024a). They are sequence-to-sequence generative models, where the sequence length is typically denoted by $n$. The key architectural component that distinguishes transformers from earlier models is the attention mechanism, which computes a softmax over the pairwise interactions of query-key vectors. However, computing the full softmax distribution requires $\Omega(n^2)$ time, due to the size of the attention matrix. This quadratic dependency renders transformers inefficient for long sequences, motivating a rich body of work aimed at approximating attention in subquadratic time. These approaches can be broadly categorized into three main classes: (1) *Pattern-based sparse attention*: only a subset of attention matrix entries are computed, with the subset determined by predefined patterns, such as sliding windows or graph-based sparsity structures (Daras et al., 2020; Kitaev et al., 2020; Roy et al., 2021; Sun et al., 2022; Child et al., 2019; Beltagy et al., 2020; Ainslie et al., 2020; Zaheer et al., 2020). (2) *Kernel-based linear attention*: these methods attempt to linearize the kernel by exploiting the identity $\mathsf{K}(x_i, x_j) = \langle \phi(x_i), \phi(x_j) \rangle$ for a feature map $\phi : \mathbb{R}^d \to \mathbb{R}^m$. When the kernel is exponential, exact computation requires $m = \infty$, so many heuristic approximations for $\phi$ have been proposed (Katharopoulos et al., 2020; Choromanski et al., 2021; Wang et al., 2020; Peng et al., 2021) with $m = O(d)$. (3) *Data structure-based attention*: these works design specialized data structures for approximating various components of attention. Examples include estimating the normalization factor via kernel density estimation (KDE) (Zandieh et al., 2023), using hashing to identify large entries (Han et al., 2024), applying polynomial approximation methods under bounded input conditions (Alman & Song, 2023), and other algorithmic innovations (Kacham et al., 2024; Zandieh et al., 2024; van den Brand et al., 2024; Song et al., 2024; Kannan et al., 2025; Chu et al., 2024; Chen et al., 2025b; Indyk et al., 2025). Our work falls into the third category, as we design quantum data structures to approximate each of the matrices involved in the attention computation.

**Quantum Machine Learning.** Given a machine learning problem, can we solve it faster on a quantum computer? The paradigm of using quantum mechanics to accelerate machine learning algorithms has sparked significant interest, leading to a wide array of results across diverse problem domains, including clustering (Kerenidis et al., 2019; Xue et al., 2023), classification (Li et al., 2019), regression (Chen & de Wolf, 2023), training neural networks (Chakrabarti et al., 2019; Kerenidis et al., 2020), convex optimization (Chakrabarti et al., 2020; van Apeldoorn et al., 2020a; Li & Zhang, 2022; Sidford & Zhang, 2023; Zhang et al., 2024b; Wang et al., 2024), mathematical programming (Brandão et al., 2019; van Apeldoorn et al., 2020b; van Apeldoorn & Gilyén, 2019; Kerenidis & Prakash, 2020; Kerenidis et al., 2021; van Apeldoorn et al., 2021; Apers & Gribling, 2023), graph sparsification (Apers & De Wolf, 2022), and recommender systems (Kerenidis & Prakash, 2017). Among the key quantum techniques, Grover search (Grover, 1996) plays a foundational role. It provides a quadratic speedup for database search problems: given a function $f : [n] \to \{0, 1\}$, the goal is to list up to $m$ indices $i$ such that $f(i) = 1$. The Grover search algorithm requires oracle access to $f$ and can produce these $m$ indices using only $O(\sqrt{mn})$ queries, in contrast to the $O(n)$ queries required classically. Several variants of Grover search have been developed to suit different computational settings. In this paper, we use the probabilistic version: given a list of $n$ probabilities $p_1, \ldots, p_n \in [0, 1]$, Grover search can be used to sample a list of indices where each $i$ is selected independently with probability $p_i$. By the standard analysis of Grover search, this sampling requires $\widetilde{O}(\sqrt{nP})$ queries to the probability values $p_i$ where $P = \sum_{i=1}^n p_i$. Before our work, Gao et al. (2023) also applied Grover search to accelerate attention computation. However, their method requires a structural assumption: for each query $q_i \in \mathbb{R}^d$, the associated set $S_i = \{j \in [n] : \langle q_i, k_j \rangle \geq \tau\}$ must

have cardinality at most $k$. Under this assumption, their algorithm runs in time $\widetilde{O}(n^{1.5}k^{0.5}d + nkd)$. Notably, if $k = n$, then their algorithm offers no speedup over the exact computation.

## 5 CONCLUSION

We consider the problem of approximating the attention module in the row query model, where the goal is to return individual rows of the approximate attention matrix. We design a quantum data structure that preprocesses $Q$, $K$, and $V$ in $\widetilde{O}(\epsilon^{-1}n^{0.5}\operatorname{poly}(s_\lambda, d, \alpha))$ time, and answers any row query in $\widetilde{O}(s_\lambda^2 + s_\lambda d)$ time. To the best of our knowledge, this is the first quantum algorithm to achieve sublinear dependence on $n$ even in the row query model.

Our work also has several limitations, which raises interesting open questions. (1) The error guarantee we obtain is in Frobenius norm rather than spectral norm. While Frobenius norm bounds the sum of the squared $\ell_2$ errors across all rows, the spectral norm provides a worst-case guarantee that each row is well approximated. Therefore, it would be desirable to strengthen the result to achieve a spectral norm guarantee. (2) The error bound we obtain contains an additive $\lambda\sqrt{n}$ term, which stems from bounding the infinity norm of the error matrix by $\sqrt{n}$ times the spectral norm of it. This bound seems overly pessimistic, and it theoretically forces one to choose small value for $\lambda$, hindering the advantage of small statistical dimension. It would be interesting to remove the $\sqrt{n}$ factor in the additive term. (3) While we provide a preliminary bit complexity analysis of our algorithm in Section F, we feel a more comprehensive study of bit complexity of numerical linear algebra in the QRAM model is needed. We leave this as a major future direction, as it will significantly broaden the practicality of these quantum speedups. (4) Our algorithm in its current form can only compute the full attention *without* the causal mask, as using the Nyström approximation implicitly assumes the complete interactions between queries and keys. To implement causal masking, one possibility is to design a quantum kernel density estimation data structure as shown in Zandieh et al. (2023) classically.

### ETHICS STATEMENT

Our work is a theoretical quantum framework to approximate the attention module in sublinear time. We don't foresee any potential ethics concerns.

### REPRODUCIBILITY STATEMENT

We include all the proofs in the appendix. For proofs of the exponential kernel, see Section A, for proofs of estimating the normalization factor, see Section B. For proofs of the leverage score approximate matrix multiplication, see Section C. The final conclusion is proved in Section D. We provide empirical verifications on the assumptions of the parameters in Section E, and a preliminary bit complexity analysis of our algorithm in Section F.

### ACKNOWLEDGMENT

We would like to thank anonymous for very helpful discussions, and Ruizhe Zhang for answering our questions on the QRAM model. Lichen Zhang is supported by a Mathworks Fellowship and a Simons Dissertation Fellowship in Mathematics.

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

# Appendix

**Roadmap.** In Section A, we describe the quantum algorithm for exponential kernels. In Section B, we discuss how to estimate the normalization factor. In Section C, we show the details on approximating matrix multiplication via leverage scores. In Section D, we combine things together and obtain the main result. In Section E, we empirically verify the assumptions on the parameters. In Section F, we discuss the bit complexity of our algorithm.

## A  QUANTUM ALGORITHM FOR EXPONENTIAL KERNEL

In this section, we give a generic reduction from attention matrix to a kernel matrix. Given queries and keys $Q = \{q_1, \ldots, q_n\}$, $K = \{k_1, \ldots, k_n\}$, recall that we are interested in the matrix $\exp(QK^\top)$ where the $(i, j)$-th entry is $\exp(q_i^\top k_j)$, and this matrix is not a PSD kernel matrix. We show a reduction that first computes the exponential kernel $\mathsf{K}(x, y) = \exp(\langle x, y \rangle)$ over the dataset $Q \cup K$, then we can effectively extract certain blocks of the kernel matrix $E$ that approximates $\exp(QK^\top)$ well. We start with a lemma on block approximation.

**Lemma A.1.** *Let $E \in \mathbb{R}^{2n \times 2n}$ be a PSD matrix and $E = \begin{bmatrix} B & A \\ A^\top & C \end{bmatrix}$ where each block is of size $n \times n$. Suppose there exists a matrix $\widetilde{E} \in \mathbb{R}^{2n \times 2n}$ such that $E \preceq \widetilde{E} \preceq E + \lambda I$ for $\lambda > 0$ and let $\widetilde{E} = \begin{bmatrix} \widetilde{B} & \widetilde{A} \\ \widetilde{A}^\top & \widetilde{C} \end{bmatrix}$, then we have*

$$\|A - \widetilde{A}\| \leq \lambda \text{ and } \|A - \widetilde{A}\|_F \leq \lambda\sqrt{n}.$$

*Proof.* We let $v \in \mathbb{R}^n$ be the vector that realizes the spectral norm $A - \widetilde{A}$, consider the augmented vector $\begin{bmatrix} \mathbf{0}_n \\ v \end{bmatrix}$, then we see that

$$\left\| (E - \widetilde{E}) \begin{bmatrix} \mathbf{0}_n \\ v \end{bmatrix} \right\|_2^2 = \left\| \begin{bmatrix} (A - \widetilde{A})v \\ (C - \widetilde{C})v \end{bmatrix} \right\|_2^2$$
$$= \|(A - \widetilde{A})v\|_2^2 + \|(C - \widetilde{C})v\|_2^2$$
$$\leq \lambda^2,$$

where the last step is by $\|E - \widetilde{E}\| \leq \lambda$ and our test vector is unit norm. As $\|(C - \widetilde{C})v\|_2^2$ is trivially non-negative, we conclude that $\|(A - \widetilde{A})v\|_2 = \|A - \widetilde{A}\| \leq \lambda$, as desired. To obtain a Frobenius norm bound, note that $\|A - \widetilde{A}\|_F \leq \sqrt{n} \cdot \|A - \widetilde{A}\| \leq \lambda\sqrt{n}$. $\qquad\square$

Our plan is to form the kernel matrix over the dataset $Q \cup K$ implicitly via Nyström approximation, then extract corresponding blocks to approximate $\exp(QK^\top)$.

**Corollary A.2.** *Let $Q, K \in \mathbb{R}^{n \times d}$ and let $E \in \mathbb{R}^{2n \times 2n}$ be the exponential kernel matrix over the dataset $Q \cup K$, suppose there exists an $\widetilde{E} \in \mathbb{R}^{2n \times 2n}$ such that $E \preceq \widetilde{E} \preceq E + \lambda I$ for some $\lambda > 0$, then there exists $\widetilde{A} \in \mathbb{R}^{n \times n}$ such that*

$$\|\widetilde{A} - \exp(QK^\top)\| \leq \lambda \text{ and } \|\widetilde{A} - \exp(QK^\top)\|_F \leq \lambda\sqrt{n}.$$

*Proof.* The result is a consequence of Lemma A.1 by identifying that

$$E = \begin{bmatrix} \exp(QQ^\top) & \exp(QK^\top) \\ \exp(KQ^\top) & \exp(KK^\top) \end{bmatrix},$$

and $\widetilde{E}$ contains proper approximations for the desired blocks. $\qquad\square$

It remains to give an efficient algorithm to approximate the exponential kernel matrix $E$. A popular scheme is via Nyström approximation (Williams & Seeger, 2000): the algorithm selects a subset of

"landmark" points, and constructs $\widetilde{E}$ through these landmarks. Musco & Musco (2017) uses recursive ridge leverage score sampling to generate such an approximation efficiently. Musco & Musco (2017) presents an algorithm that uses $\widetilde{O}(ns_\lambda \log(1/\delta))$ kernel function evaluations and $\widetilde{O}(ns_\lambda^2 \log(1/\delta))$ additional runtime to compute an approximation $\widetilde{K}$ satisfying $K \preceq \widetilde{K} \preceq K + \lambda I$ with probability at least $1 - \delta$. We restate their main result here for the sake of completeness.

**Lemma A.3** (Theorem 7 of Musco & Musco (2017)). *Let $s = O(s_\lambda \log(s_\lambda/\delta))$, there exists a weighted sampling matrix $S \in \mathbb{R}^{n \times s}$, such that the Nyström approximation of $E$, $\widetilde{E} = ES(S^\top ES)^\dagger S^\top E$ satisfies*

$$E \preceq \widetilde{E} \preceq E + \lambda I,$$

*holds with probability at least $1 - \delta$. Moreover, $S$ can be computed using $O(ns)$ kernel evaluations and $O(ns^2)$ additional time.*

Our main contribution is a quantum algorithm that generates the approximation in *sublinear time*. Before introducing the algorithm, we recall several key concepts.

Lemma A.3 relies on approximating the ridge leverage score on a sample, which can be captured by the notion of generalized ridge leverage score.

**Definition A.4** (Generalized ridge leverage score, Musco & Musco (2017)). *Let $E \in \mathbb{R}^{n \times n}$ be a kernel matrix, let $\lambda > 0$, and let $S \in \mathbb{R}^{n \times s}$ be any weighted sampling matrix, the $\lambda$-generalized ridge leverage score with respect to $S$, is defined for any $i \in [n]$,*

$$\widetilde{\tau}_i^\lambda := \frac{1}{\lambda}(E - ES(S^\top ES + \lambda I)^{-1} S^\top E)_{i,i},$$

*let $B \in \mathbb{R}^{n \times n}$ be any factorization of $E = BB^\top$, it can be equivalently defined as*

$$\widetilde{\tau}_i^\lambda = b_i^\top (B^\top S^\top SB + \lambda I)^{-1} b_i,$$

*where $b_i$ is the $i$-th row of $B$.*

We also need a procedure introduced in Apers & Gribling (2023) that generates a spectral approximation of an $n \times d$ matrix, given only queries to its rows, using quantum leverage score sampling. We record it here.

**Lemma A.5** (Theorem 3.1 of Apers & Gribling (2023)). *Let $U \in \mathbb{R}^{n \times d}$, $\epsilon, \delta \in (0, 1)$. There exists a quantum algorithm that computes a weighted sampling matrix $S \in \mathbb{R}^{n \times s}$ with $s = O(\epsilon^{-2} d \log(d/\delta))$ such that with probability at least $1 - \delta$,*

$$(1 - \epsilon)U^\top U \preceq U^\top SS^\top U \preceq (1 + \epsilon)U^\top U.$$

*The quantum algorithm uses $\widetilde{O}(\epsilon^{-1} n^{0.5} d^{0.5})$ row queries to $U$, and it takes time $\widetilde{O}(\epsilon^{-1} n^{0.5} d^{1.5} + d^\omega)$. Moreover, if the leverage score sampling matrix contains $s \leq d$ rows, then the algorithm uses $\widetilde{O}(n^{0.5} s^{0.5})$ row queries to $U$ and it takes time $\widetilde{O}(n^{0.5} s^{0.5} d + d^\omega)$. We use $\mathrm{QLEVERAGESCORE}(U, s)$ to denote this procedure that produces a leverage score sampling matrix $S \in \mathbb{R}^{n \times s}$.*

We prove the key algorithmic result of this section.

**Theorem A.6.** *Let $\{x_1, \ldots, x_n\} \subseteq \mathbb{R}^d$ be a dataset, $\mathsf{K} : \mathbb{R}^d \times \mathbb{R}^d \to \mathbb{R}^m$ be a kernel function, $\lambda > 0$ and $\delta \in (0, 1)$. Let $E$ be the kernel matrix where $E_{i,j} = \mathsf{K}(x_i, x_j)$. Suppose $s = O(s_\lambda \log(s_\lambda/\delta))$, then Algorithm 1 computes a weighted sampling matrix $S \in \mathbb{R}^{n \times s}$ such that with probability at least $1 - \delta$,*

$$E \preceq \widetilde{E} \preceq E + \lambda I,$$

*where $\widetilde{E} = ES(S^\top ES)^\dagger S^\top E$. Moreover, $S$ can be computed in $\widetilde{O}(n^{0.5} s^{0.5})$ row queries to $Q, K$ and in time $\widetilde{O}(n^{0.5} s^{1.5} \cdot (\mathcal{T}_\mathsf{K} + s) + s^\omega)$, where $\mathcal{T}_\mathsf{K}$ is the time to evaluate the kernel function.*

*Proof.* We note that the major differences between Algorithm 1 and the algorithm in Musco & Musco (2017) are

- Musco & Musco (2017) algorithm is recursive, our algorithm unrolls the recursion and iteratively constructs the weighted sampling matrix;

- Musco & Musco (2017) computes all $p_i$'s classically, while we use QSAMPLE to generate samples.

Hence, the correctness is automatically satisfied. It remains to give a bound on the running time.

- Computing $M_0$: $M_0 \in \mathbb{R}^{s \times s}$ contains the values of kernel functions over $s^2$ pairs, forming it takes $O(s^2) \cdot \mathcal{T}_{\mathsf{K}}$ time;

- Computing $\widehat{M}$: we maintain the invariant that $M_t \in \mathbb{R}^{s \times s}$ for all $t \in [T]$, therefore computing $\widehat{M}$ is inverting an $s \times s$ matrix, which takes $O(s^\omega)$ time;

- Computing $D_{t-1}^\top K_i$: this operation involves computing $s$ weighted kernel function evaluations, given $D_{t-1}$ stores a list of $s$ indices together with weights, it can be done in $O(s) \cdot \mathcal{T}_{\mathsf{K}}$ time;

- Oracle for $q_i$: for any fixed $i$, note that we need to form $D_{t-1}^\top K_i$ using $O(s) \cdot \mathcal{T}_{\mathsf{K}}$ time, and computing the quadratic form takes $O(s^2)$ time. Thus each oracle call takes $O(s) \cdot \mathcal{T}_{\mathsf{K}} + O(s^2)$ time;

- Computing $\widetilde{D}_t$: this step requires to compute at most $n$ probabilities, and each probability can be computed via an oracle call in $O(s) \cdot \mathcal{T}_{\mathsf{K}} + O(s^2)$ time, so it remains to give a bound on the sum of probabilities. By the definition of $p_i$,

$$\sum_{i=1}^n p_i \le 16 \log(2s/\delta) \sum_{i=1}^n q_i,$$

and the sum of $q_i$'s is

$$\sum_{i=1}^n q_i = \frac{5}{\lambda} \cdot \left( \mathsf{K}(x_i, x_i) - (D_{t-1}^\top K_i)^\top \widehat{M} (D_{t-1}^\top K_i) \right)$$

$$= \frac{5}{\lambda} \cdot (E - E D_{t-1}(D_{t-1}^\top E D_{t-1} + \lambda I)^{-1} D_{t-1}^\top E)_{i,i}$$

$$= 5 \cdot \sum_{i=1}^n \widetilde{\tau}_i^\lambda,$$

by Theorem 8 of Musco & Musco (2017), the sum of $\lambda$-generalized ridge leverage score with sampling matrix $D_{t-1}$ is at most $O(s_\lambda \log(s_\lambda/\delta)) = s$, thus the runtime is $\widetilde{O}(n^{0.5} s^{1.5} \cdot (\mathcal{T}_{\mathsf{K}} + s))$.

Finally, note that the loop is dominated by the last iteration, and at each iteration, the number of points to consider is divided by half, we conclude the overall runtime of Algorithm 1 is

$$\widetilde{O}(n^{0.5} s^{1.5} \cdot (\mathcal{T}_{\mathsf{K}} + s) + s^\omega),$$

as desired. $\qquad \square$

We can then apply Theorem A.6 to exponential kernel function and the dataset $Q \cup K$ to compute a Nyström sampling matrix $S$.

**Corollary A.7.** *Let $Q, K \in \mathbb{R}^{n \times d}$, $\lambda > 0$ and $\delta \in (0, 1)$. Define the dataset $X = \{x_1, x_2, \ldots, x_{2n}\} \subseteq \mathbb{R}^d$ where for $i \in [n]$, $x_i = q_i$ and for $i \in \{n+1, \ldots, 2n\}$, $x_i = k_i$. Let $E$ be the kernel matrix where $E_{i,j} = \exp(\langle x_i, x_j \rangle)$. Suppose $s = O(s_\lambda \log(s_\lambda/\delta))$, then there exists an algorithm that computes a weighted sampling matrix $S \in \mathbb{R}^{2n \times s}$ such that, let $\widetilde{E} = E S(S^\top E S)^\dagger S^\top E$, then with probability at least $1 - \delta$, $E \preceq \widetilde{E} \preceq E + \lambda I$. Moreover, $S$ can be computed in $\widetilde{O}(n^{0.5} s^{1.5} \cdot (d + s) + s^\omega)$ time.*

*Proof.* Apply Theorem A.6 to the kernel function $\mathsf{K}(x_i, x_j) = \exp(\langle x_i, x_j \rangle)$ and note that the kernel function can be computed in $O(d)$ time. $\qquad \square$

## B  ESTIMATING THE NORMALIZATION FACTOR

Given a sublinear quantum algorithm to approximate the matrix $\exp(QK^\top)$, our next step is to estimate the normalization factor $\exp(QK^\top)\mathbf{1}_n$ to compute the softmax matrix. We first show that given a Nyström approximation to the $2n \times 2n$ kernel matrix $E$, how to compute the normalization factor and the approximate guarantees.

**Lemma B.1.** *Let $M \in \mathbb{R}^{n \times n}$ be a symmetric matrix, then we have*

$$\|M\|_\infty \leq \sqrt{n} \cdot \|M\|.$$

*Proof.* Fix any $i \in [n]$, we examine the row $M_{i,*}$, set the test vector $x$ to be $x_j = \begin{cases} +1, & \text{if } M_{i,j} \geq 0, \\ -1, & \text{otherwise.} \end{cases}$, then

$$\begin{aligned}
\|M_{i,*}\|_1 &= M_{i,*}^\top x \\
&= \langle Me_i, x \rangle \\
&\leq \|Me_i\|_2 \cdot \|x\|_2 \\
&\leq \|M\| \cdot \|x\|_2 \\
&= \sqrt{n} \cdot \|M\|.
\end{aligned}$$

The conclusion can be achieved by noting that this bound works for any row $i$. $\qquad \square$

There are two major issues for estimating the normalization factor:

- Corollary A.7 only allows us to compute the sampling matrix in sublinear time, explicitly forming the Nyström approximation $\widetilde{E}$ however, would require $\Omega(n)$ time since the matrix is of size $n \times n$;

- Even though we are given the explicit factorization $\widetilde{E} = UU^\top$ where $U \in \mathbb{R}^{2n \times s}$, we would have to compute $n$ normalization factors, which would require $\Omega(n)$ time.

In other words, because the output has size $\Omega(n)$, one cannot expect any quantum algorithm to run in $o(n)$ time. Instead, we design a quantum data structure with preprocessing time $o(n)$ time, and can support query to compute the normalization factor to any row efficiently.

In particular, we are interested in the following algorithmic task: given query access to the rows of a matrix $U \in \mathbb{R}^{n \times s}$ and a vector $v \in \mathbb{R}^n$, output a vector $\widetilde{\mu} \in \mathbb{R}^s$ such that $\|\widetilde{\mu} - U^\top v\|_{(U^\top U)^{-1}} \leq \epsilon$, which can be solved via Lemma 2.6. For our application, $\|v\|_\infty = 1$. However, we are interested in the quantity $UU^\top v$ so we need to measure the error $\|U(\widetilde{\mu} - U^\top v)\|_2$. How would a bound on the $\|\cdot\|_{(U^\top U)^{-1}}$ be useful? We prove a structural lemma below.

**Lemma B.2.** *Let $x \in \mathbb{R}^s$ and $U_2 \in \mathbb{R}^{n \times s}$ satisfying $\|x\|_{(U_2^\top U_2)^{-1}} \leq \epsilon$ for some $\epsilon \in (0, 1)$, let $U_1 \in \mathbb{R}^{n \times s}$, we have*

$$\|U_1 x\|_2 \leq \epsilon \cdot \|U_1 U_2^\top\|.$$

*Proof.* We define the vector $y = (U_2^\top U_2)^{-1} x$ and $z = U_2 y$, then

$$\begin{aligned}
\|z\|_2^2 &= y^\top U_2^\top U_2 y \\
&= x^\top (U_2^\top U_2)^{-1} x \\
&= \|x\|_{(U_2^\top U_2)^{-1}}^2 \\
&\leq \epsilon^2,
\end{aligned}$$

moreover, the vector of interest is $U_1 x$ which is

$$\begin{aligned}
U_1 x &= U_1 (U_2^\top U_2) y \\
&= (U_1 U_2^\top) U_2 y
\end{aligned}$$

$$= (U_1 U_2^\top) z,$$

subsequently its $\ell_2$ norm can be bounded as

$$\|U_1 x\|_2 \leq \|U_1 U_2^\top\| \cdot \|z\|_2$$
$$\leq \epsilon \cdot \|U_1 U_2^\top\|,$$

as desired. $\qquad\square$

**Corollary B.3.** *Let $\epsilon \in (0,1), U_1, U_2 \in \mathbb{R}^{n \times s}$ where $\widetilde{A} = U_1 U_2^\top$, $v \in \mathbb{R}^n$, suppose there exists a vector $\widetilde{\mu} \in \mathbb{R}^s$ with $\|\widetilde{\mu} - U_2^\top v\|_{(U_2^\top U_2)^{-1}} \leq \epsilon$, then we have*

$$\|\widetilde{A} v - U_1 \widetilde{\mu}\|_2 \leq \epsilon \cdot \|U_1 U_2^\top\|.$$

*Proof.* We will apply Lemma B.2 by setting $x = \widetilde{\mu} - U_2^\top v$ and by noting that $U_1 x = U_1 \widetilde{\mu} - U_1 U_2^\top v = U_1 \widetilde{\mu} - \widetilde{A} v$. $\qquad\square$

We are now in the position to state our formal theorem, which provides an end-to-end guarantee on estimating the normalization factor. For simplicity, we will prove the statement with high probability guarantee, i.e., the success probability is $1 - 1/\operatorname{poly}(n)$.

**Theorem B.4.** *Let $Q, K \in \mathbb{R}^{n \times d}$, $\lambda > 0$ and $\epsilon \in (0,1)$. Let $s = \widetilde{O}(s_\lambda)$ where $s_\lambda$ is the statistical dimension of the exponential kernel on $Q \cup K$. There exists a data structure (Algorithm 2) with the following specification:*

- *Preprocessing in $\widetilde{O}(n^{0.5} s^{0.5}/\epsilon)$ row queries to $Q, K$ and time $\widetilde{O}(n^{0.5} s^{1.5}(s+d)/\epsilon + s^\omega)$;*

- *For any $i \in [n]$, it outputs an approximate normalization factor for row $i$ in time $O(s(s+d))$.*

*Moreover, with probability at least $1 - 1/\operatorname{poly}(n)$, it holds that for any $i \in [n]$, the output $b_i$ satisfies*

$$|b_i - \exp(q_i K^\top) \mathbf{1}_n| \leq O(\epsilon \|A\| + \lambda \sqrt{n}),$$

*if $\frac{\lambda \sqrt{n}}{\|A\|} \leq 1$, then the bound can be further simplified to*

$$|b_i - \exp(q_i K^\top) \mathbf{1}_n| \leq O(\lambda \sqrt{n}),$$

*and the preprocessing time simplifies to*

$$\widetilde{O}(s^{1.5}(s+d)\|A\|/\lambda + s^\omega).$$

*Proof.* Given $Q, K$, let $E \in \mathbb{R}^{2n \times 2n}$ be the associated exponential kernel matrix. We will first invoke Corollary A.7 to compute a sampling matrix $S \in \mathbb{R}^{2n \times s}$ where $s = \widetilde{O}(s_\lambda)$ such that $\widetilde{E} = ES(S^\top ES)^\dagger S^\top E$ approximates $E$, in time $\widetilde{O}(n^{0.5} s^{1.5}(s+d) + s^\omega)$. Set $U = ES(S^\top ES)^{\dagger/2}$, we have that $\widetilde{E} = UU^\top$ for $U = \begin{bmatrix} U_1 \\ U_2 \end{bmatrix}$ with $U_1, U_2 \in \mathbb{R}^{n \times s}$, and our desired approximate block for $A$ is $\widetilde{A} = U_1 U_2^\top$. Note that forming $U$ explicitly would take $\Omega(n)$ time, so we instead implement a row oracle for $U_2$. Since $U_2 \in \mathbb{R}^{n \times s}$, we only need to compute $s$ entries for each row, and let $N = (S^\top ES)^{\dagger/2}$, we see that $(U_2)_{j,*} = N(ES)_{j+n,*}$ and $(ES)_{j+n,*}$ contains values in the form of $S_k \cdot \exp(\langle x_{j+n}, x_k \rangle)$ for $k \in S$. $N$ can be computed in $O(s^2 d + s^\omega)$ time, and row oracle for any $j \in [n]$ can be implemented in $O(sd + s^2)$ time. By Lemma 2.6, $\widetilde{\mu}$ can be computed in $\widetilde{O}(n^{0.5} s^{1.5}(s+d)/\epsilon)$ time. To query the normalization factor for row $i$, note that it can be computed via $(U_1 \widetilde{\mu})_i = \langle (U_1)_{i,*}, \widetilde{\mu} \rangle$, which can be computed using row oracle, in $O(s(s+d))$ time. Thus, the overall runtime of our procedure can be summarized as

- Preprocessing time $\widetilde{O}(n^{0.5} s^{1.5}(s+d)/\epsilon + s^\omega)$;

- Query time $O(s(s+d))$.

It remains to give an approximation guarantee. With probability at least $1 - 1/\operatorname{poly}(n)$, we have $\|A - \widetilde{A}\| \leq \lambda$, and observe that

$$
\begin{aligned}
|\widetilde{a}_i^\top \mathbf{1_n} - \exp(q_i K^\top) \mathbf{1}_n| &\leq \|(\widetilde{A} - A)v\|_\infty \\
&\leq \|\widetilde{E} - E\|_\infty \cdot \|v\|_\infty \\
&\leq \lambda \sqrt{n},
\end{aligned}
$$

where the second step is by the matrix infinity norm is the induced norm of vector $\ell_\infty$ norm, and the last step is by Lemma B.1. On the other hand, our final output $b_i$ is an approximation to $\widetilde{a}_i^\top \mathbf{1}_n$. Let $\widetilde{y} := U_1 \widetilde{\mu}$, by Corollary B.3, we have

$$
\|\widetilde{A}v - \widetilde{y}\|_2 \leq \epsilon \cdot \|\widetilde{A}\|,
$$

this holds with probability at least $1 - \delta$, conditioning on this event, note that by Lemma A.1, we have that $\|\widetilde{A}\| \leq \|A\| + \lambda$. Thus, we conclude our final result by

$$
\begin{aligned}
|b_i - \exp(q_i K^\top) \mathbf{1}_n| &\leq |b_i - \widetilde{a}_i \mathbf{1}_n| + |\widetilde{a}_i^\top \mathbf{1}_n - \exp(q_i K^\top) \mathbf{1}_n| \\
&\leq \|\widetilde{A}v - \widetilde{y}\|_2 + \lambda \sqrt{n} \\
&\leq \epsilon \cdot (\lambda + \|A\|) + \lambda \sqrt{n}.
\end{aligned}
$$

Now, suppose $\lambda \sqrt{n} \leq \|A\|$, then we could set $\epsilon = \frac{\lambda \sqrt{n}}{\|A\|}$, the error bound simplifies to $O(\lambda \sqrt{n})$. □

## C APPROXIMATE MATRIX MULTIPLICATION VIA LEVERAGE SCORE

It remains to handle the value matrix, and we will do so via a machinery called approximate matrix multiplication.

**Definition C.1** (Approximate matrix multiplication, Clarkson & Woodruff (2017)). *Let $A \in \mathbb{R}^{n \times d}, B \in \mathbb{R}^{n \times m}$ and let $C = A^\top B \in \mathbb{R}^{d \times m}$. The approximate matrix multiplication problem asks to design a random matrix $S \in \mathbb{R}^{n \times s}$, such that*

$$
\Pr[\|A^\top S S^\top B - C\|_F \leq \epsilon \|A\|_F \|B\|_F] \geq 1 - \delta,
$$

*where $\epsilon, \delta \in (0, 1)$. We call such $S$ satisfying $(\epsilon, \delta)$-AMM.*

To generate the random matrix $S$, our strategy will be performing leverage score sampling over $V$. However, standard proof (see, e.g., Clarkson & Woodruff (2017)) requires $V$ to have orthonormal columns. We provide a proof for the case where $V$ does not have orthonormal columns (albeit it requires extra factors in blowups). Before doing so, we define a parameter that quantifies this blowup which we call *row distortion*.

**Definition C.2** (Row distortion). *Let $A \in \mathbb{R}^{n \times d}$ for $n \geq d$, we define the row distortion of $A$, denoted by $\alpha(A)$, as*

$$
\alpha(A) := \frac{d}{\|A\|_F^2} \cdot \max_{i \in [n]} \frac{\|a_i\|_2^2}{\tau_i},
$$

*where $a_i$ is the $i$-th row of $A$ and $\tau_i$ is the $i$-th leverage score of $A$ (Definition 2.1). When $A$ is clear from context, we use $\alpha$ as an abbreviation.*

**Lemma C.3.** *Let $A \in \mathbb{R}^{n \times d}$ with $n \geq d$, then the row distortion of $A$ satisfies*

$$
\alpha(A) \leq \frac{d}{\operatorname{srank}(A)},
$$

*where $\operatorname{srank}(A) = \frac{\|A\|_F^2}{\|A\|^2}$ is the stable rank of $A$.*

*Proof.* We derive an upper bound on $\|a_i\|_2^2$, let $A = U\Sigma V^\top$ be its SVD, then

$$
\|a_i\|_2^2 = \|e_i^\top U \Sigma V^\top\|_2^2
$$

$$\leq \|u_i\|_2^2 \cdot \|\Sigma V^\top\|^2$$
$$= \tau_i \cdot \|U\Sigma V^\top\|^2$$
$$= \tau_i \cdot \|A\|^2,$$

where the third step is by the definition of leverage score and spectral norm is unitary invariant. We thus obtain the following bound on $\alpha(A)$:

$$\alpha(A) = \frac{d}{\|A\|_F^2} \cdot \max_{i\in[n]} \frac{\|a_i\|_2^2}{\tau_i}$$
$$\leq \frac{d}{\|A\|_F^2} \cdot \max_{i\in[n]} \frac{\tau_i \cdot \|A\|^2}{\tau_i}$$
$$= d \cdot \frac{\|A\|^2}{\|A\|_F^2}$$
$$= \frac{d}{\mathrm{srank}(A)},$$

where we recall that $\mathrm{srank}(A) = \frac{\|A\|_F^2}{\|A\|^2}$. $\qquad\square$

We are now ready to prove a generalized approximate matrix multiplication based on leverage score sampling, when the matrix does not have orthonormal columns.

**Lemma C.4.** *Let $A \in \mathbb{R}^{n\times d}, B \in \mathbb{R}^{n\times m}$, let $S \in \mathbb{R}^{n\times s}$ be the leverage score sampling matrix of $A$ with $s = (\epsilon^{-2}\alpha \log(1/\delta))$ for $\epsilon, \delta \in (0,1)$ and $\alpha$ is the row distortion of $A$ (Definition C.2). Then, $S$ is an $(\epsilon, \delta)$-AMM.*

*Proof.* For the sampling matrix $S$, it is a scaled submatrix of the permutation matrix, where for any $m \in [s]$, $S_{m,z_m} = \frac{1}{\sqrt{sp_m}}$ where $p_m \geq \frac{\tau_m}{d}$ and $z_m = i$ with probability $p_i$. Let $a_i, b_j$ denote the $i$-th and $j$-th row of $A$ and $B$, respectively. We can write

$$A^\top SS^\top B - A^\top B = \frac{1}{s} \sum_{i\in[n],m\in[s]} a_i b_i^\top \left(\frac{\mathbb{I}[z_m = i]}{p_i} - 1\right),$$

taking expectation, we obtain

$$\mathbb{E}[A^\top SS^\top B - A^\top B] = \frac{1}{s} \sum_{i=1}^n a_i b_i^\top \left(\frac{p_i}{p_i} - 1\right)$$
$$= 0,$$

to bound the second moment of $\|A^\top SS^\top B - A^\top B\|_F$, we first expand the definition of Frobenius norm square:

$$\mathbb{E}\,\mathrm{tr}[(A^\top SS^\top B - A^\top B)(A^\top SS^\top B - A^\top B)]$$
$$= \mathbb{E}\frac{1}{s^2}\,\mathrm{tr}\left[\sum_{i,j\in[n],m\in[s]} b_j a_j^\top a_i b_i^\top \left(\frac{\mathbb{I}[z_m = j]}{p_j} - 1\right)\left(\frac{\mathbb{I}[z_m = i]}{p_i} - 1\right)\right]$$
$$= \frac{1}{s^2}\sum_{m=1}^s \mathrm{tr}\left[\sum_{i=1}^n \frac{1}{p_i}\cdot b_i a_i^\top a_i b_i^\top - B^\top AA^\top B\right]$$
$$= \frac{1}{s}\,\mathrm{tr}\left[\sum_{i=1}^n \frac{1}{p_i}\cdot b_i a_i^\top a_i b_i^\top - B^\top AA^\top B\right]$$
$$\leq \frac{1}{s}\left(\sum_{i=1}^n \frac{1}{p_i}\|a_i\|_2^2\|b_i\|_2^2 - \mathrm{tr}[B^\top AA^\top B]\right)$$
$$\leq \frac{1}{s}(\alpha\|A\|_F^2\|B\|_F^2 - \|A^\top B\|_F^2)$$

$$\leq \frac{\alpha}{s}\|A\|_F^2 \|B\|_F^2,$$

where the first step is by definition of $S$, the second step is by applying expectation and use $\mathbb{E}[A^\top S S^\top B - A^\top B] = 0$, the fourth step is by $\operatorname{tr}[b_i a_i^\top a_i b_i^\top] = \|a_i b_i^\top\|_F^2 \leq \|a_i\|_2^2 \|b_i\|_2^2$, the fifth step is by $p_i \geq \frac{\tau_i}{d}$, therefore

$$\frac{1}{p_i} \leq \frac{d}{\tau_i}$$
$$= \frac{\|A\|_F^2}{\|a_i\|_2^2} \cdot \frac{d}{\|A\|_F^2} \cdot \frac{\|a_i\|_2^2}{\tau_i}$$
$$\leq \alpha \cdot \frac{\|A\|_F^2}{\|a_i\|_2^2},$$

where the last step is by the definition of $\alpha$. By Chebyshev's inequality, we can choose $s = O(\alpha/\epsilon^2)$ so that the approximate matrix multiplication holds with constant probability, and one could boost the success probability to $1 - \delta$ by either taking $\log(1/\delta)$ independent copies via a Chernoff bound, or directly through Bernstein inequality. $\square$

We are ready to state our final result on approximating the value matrix $V$.

**Theorem C.5.** *Let $V \in \mathbb{R}^{n \times d}$, $\epsilon \in (0,1)$ and $\alpha$ be the row distortion of $V$. There exists a quantum algorithm that computes a weighted sampling matrix $S \in \mathbb{R}^{n \times s}$ with $s = \widetilde{O}(\epsilon^{-2}\alpha)$ such that for any fixed matrix $B \in \mathbb{R}^{n \times m}$, $S$ is an $(\epsilon, 1/\operatorname{poly}(n))$-AMM. Moreover, $S$ can be computed using $\widetilde{O}(\epsilon^{-1} n^{0.5} \alpha^{0.5})$ row queries to $V$ and $\widetilde{O}(\epsilon^{-1} n^{0.5} \alpha^{0.5} d + d^\omega)$ time.*

*Proof.* The proof is by composing Lemma A.5 and Lemma C.4, and note that for $\widetilde{O}(\epsilon^{-2}\alpha)$ rows, the sum of leverage scores is at most $\widetilde{O}(\epsilon^{-2}\alpha)$. $\square$

## D  PUT THINGS TOGETHER

We are now ready to state our final algorithm and its guarantee. Recall that, we define $D = \exp(QK^\top)\mathbf{1}_n$ and $D' = \exp(KQ^\top)\mathbf{1}_n$. We use $\widetilde{D}, \widetilde{D}'$ to denote their approximations.

We prove a simple inequality that quantifies the perturbation on the inverse.

**Lemma D.1.** *Let $C, D \in \mathbb{R}^{n \times n}$, if $D$ is nonsingular and $\|C - D\| \leq \epsilon$, and $\|D^{-1}\| < 1/\epsilon$, then $C$ is also nonsingular and $\|C^{-1}\| \leq \frac{\|D^{-1}\|}{1 - \epsilon \cdot \|D^{-1}\|}$.*

*Proof.* We will make use of Neumann series, which states that for $\|A\| < 1$, $(I - A)^{-1}$ admits the expansion

$$(I - A)^{-1} = \sum_{k=0}^{\infty} A^k,$$

this leads to a bound on the norm:

$$\|(I - A)^{-1}\| = \|\sum_{k=0}^{\infty} A^k\|$$
$$\leq \sum_{k=0}^{\infty} \|A^k\|$$
$$\leq \sum_{k=0}^{\infty} \|A\|^k$$
$$= \frac{1}{1 - \|A\|}, \tag{1}$$

now, to prove our desired bound, we write $C = D + E$ where $E$ is the perturbation, then $C = D + E = D(I + D^{-1}E)$, and we will apply Eq. (1) to $-D^{-1}E$:

$$\|D^{-1}E\| \leq \|D^{-1}\| \cdot \|E\|$$
$$= \|D^{-1}\| \cdot \|C - D\|$$
$$< 1/\epsilon \cdot \epsilon$$
$$= 1,$$

therefore

$$\|C^{-1}\| = \|(I + D^{-1}E)^{-1}D^{-1}\|$$
$$\leq \|D\| \cdot \|(I - D^{-1}E)^{-1}\|$$
$$\leq \frac{\|D^{-1}\|}{1 - \|D^{-1}E\|}$$
$$\leq \frac{\|D^{-1}\|}{1 - \|E\| \cdot \|D^{-1}\|}$$
$$\leq \frac{\|D^{-1}\|}{1 - \epsilon \cdot \|D^{-1}\|},$$

this completes the proof. $\qquad \square$

**Theorem D.2** (Formal version of Theorem 3.1). *Let $Q, K, V \in \mathbb{R}^{n \times d}$ be the query, key and value matrices for attention, let $\epsilon, \lambda > 0$. Let $E \in \mathbb{R}^{2n \times 2n}$ be the exponential kernel matrix with the dataset $Q \cup K$, and let $s_\lambda$ be the statistical dimension of $E$ (Definition 2.2), $\alpha$ be the row distortion of $V$ (Definition C.2). There exists a quantum data structure (Algorithm 3) that preprocesses $Q, K, V$ only through row queries to these matrices and with probability at least $1 - 1/\operatorname{poly}(n)$, for any $i \in [n]$, it outputs a vector $\widetilde{r}_i \in \mathbb{R}^d$ where*

$$\widetilde{r}_i = e_i^\top \widetilde{D}^{-1} \widetilde{A} \widetilde{V}.$$

*If in addition, we have $\|D^{-1}\| < \frac{1}{\epsilon\|A\| + \lambda\sqrt{n}}$, then the approximations $\widetilde{D}, \widetilde{A}$ and $\widetilde{V}$ satisfy that*

$$\|\widetilde{D}^{-1}\widetilde{A}\widetilde{V} - D^{-1}AV\|_F \leq \epsilon \cdot (\beta \cdot \|D^{-1}\|) \cdot (\|A\|_F + \lambda\sqrt{n}) \cdot \|V\|_F,$$

*where $\beta = \frac{1}{1 - (\epsilon\|A\| + \lambda\sqrt{n})\|D^{-1}\|}$. Moreover, the algorithm has the following runtime specification:*

- *Preprocesses in $\widetilde{O}(\epsilon^{-1}n^{0.5}s_\lambda^{0.5})$ row queries to $Q, K$ and $\widetilde{O}(\epsilon^{-1}n^{0.5}\alpha^{0.5})$ row queries to $V$, and $\widetilde{O}(\epsilon^{-1}n^{0.5}(s_\lambda^{2.5} + s_\lambda^{1.5}d + \alpha^{0.5}d) + d^\omega + s_\lambda^\omega + \epsilon^{-2}s_\lambda\alpha d)$ time;*

- *For any $i \in [n]$, it outputs $\widetilde{r}_i$ in $\widetilde{O}(s_\lambda^2 + s_\lambda d)$ time.*

*Proof.* By Theorem C.5, we know that with probability at least $1 - 1/\operatorname{poly}(n)$, the following bound holds:

$$\|\widetilde{D}^{-1}\widetilde{A}S_V S_V^\top V\|_F \leq \epsilon \cdot \|\widetilde{D}^{-1}\widetilde{A}\|_F \cdot \|V\|_F$$
$$\leq \epsilon \cdot \|\widetilde{D}^{-1}\| \cdot \|\widetilde{A}\|_F \cdot \|V\|_F,$$

where the second step is by $\|\widetilde{D}^{-1}\widetilde{A}\|_F \leq \|\widetilde{D}^{-1}\| \cdot \|\widetilde{A}\|_F$. By Theorem B.4, we know that

$$\|\widetilde{D} - D\| \leq \epsilon\|A\| + \lambda\sqrt{n},$$

note that as long as the error satisfies that $\|D^{-1}\| < \frac{1}{\epsilon\|A\| + \lambda\sqrt{n}}$, then by Lemma D.1, we obtain a bound on $\|\widetilde{D}^{-1}\|$:

$$\|\widetilde{D}^{-1}\| \leq \frac{\|D^{-1}\|}{1 - (\epsilon\|A\| + \lambda\sqrt{n})\|D^{-1}\|}.$$

Finally, by Corollary A.2, we have

$$\|\widetilde{A}\|_F \leq \|A\|_F + \lambda\sqrt{n}.$$

For the runtime, it suffices to combine Corollary A.7, Theorem B.4 and Theorem C.5, and the only additional runtime term is the $\epsilon^{-2}s_\lambda\alpha d$, which is the time to form matrix $\widetilde{R}$ and $\widetilde{L}$. $\qquad \square$

## E    EMPIRICAL VERIFICATIONS ON PARAMETERS

In this section, we empirically verify the assumptions on the parameters. In particular, we focus on the following metrics:

- $\|D^{-1}\| \leq \frac{1}{\epsilon\|A\|+\lambda\sqrt{n}}$, we specifically check that what is the maximum possible $\epsilon$ so that $\|D^{-1}\| \leq \frac{1}{\epsilon\|A\|}$.

- $\frac{\|A\|_F}{\|A\|}$, this is important as our error guarantee is in terms of Frobenius norm rather than the more typical spectral norm (Zandieh et al., 2023; Han et al., 2024), we verify that this ratio is small.

- $\frac{\|V\|_F}{\|V\|}$, this is similar to the above test, we verify that this ratio is close to $\sqrt{d}$.

- $\frac{d}{\text{srank}(V)}$, this quantity serves as an upper bound of $\alpha(V)$, we verify that this quantity is a small constant rather than the upper bound $d$.

- $\frac{\|A\|_\infty}{\|A\|}$, in our error analysis, we have to pay an extra $\sqrt{n}$ factor when converting the spectral norm to matrix infinity norm, we empirically show that this ratio is a small constant rather than the $\sqrt{n}$ scaling.

To conduct our experiment, we use the `OLMo2-1B` and `OLMo2-7B` models, in particular their `stage1` pretraining checkpoints (Walsh et al., 2025). We list the model architecture in the following.

|  | **Sequence length** $n$ | **Value dimension** $d$ | **Number of layers** $L$ | **Number of heads** $H$ |
|---|---|---|---|---|
| `OLMo2-1B` | 4096 | 128 | 16 | 16 |
| `OLMo2-7B` | 4096 | 128 | 32 | 32 |

Table 2: Model architecture for `OLMo2-1B` and `OLMo2-7B`.

We compute the corresponding attention modules $D, A, V$ using the pretraining datasets for these models, with batch size 2 and 16 batches. We then compute the statistics for each head and each layer, then aggregate the statistics over all layers. We report the mean of these statistics.

|  | $\epsilon_{\max}$ | $\frac{\|A\|_F}{\|A\|}$ | $\frac{\|V\|_F}{\|V\|}$ | $\frac{d}{\text{srank}(V)}$ | $\frac{\|A\|_\infty}{\|A\|}$ |
|---|---|---|---|---|---|
| `OLMo2-1B` | 0.1708 | 1.3769 | 11.3137 | 1.7126 | 2.7439 |
| `OLMo2-7B` | 0.1685 | 1.3586 | 11.3137 | 2.1345 | 2.7439 |

Table 3: Mean statistics across all heads and all layers. $\epsilon_{\max}$ is the maximum $\epsilon$ such that $\|D^{-1}\| \leq \frac{1}{\epsilon\|A\|}$.

Through these verifications, we make the following preliminary observations:

- To satisfy the $\|D^{-1}\| \leq \frac{1}{\epsilon\|A\|}$ assumption, it is enough to pick $\epsilon \leq 0.17$, which is larger than common choice of $\epsilon \approx 0.1$. This gives us enough room to tune the parameter $\epsilon$ to achieve a good balance between efficiency and accuracy.

- The ratio $\frac{\|A\|_F}{\|A\|}$ is a constant smaller than 2, much smaller than the worst case $\sqrt{n}$ predicted by the theory (recall that $n = 4096$ and $\sqrt{n} = 64$). This suggests that we don't need to scale down $\epsilon$ by a factor of $\sqrt{n}$ to recover the spectral norm error guarantee.

- The ratio $\frac{\|V\|_F}{\|V\|} \approx 11$ is roughly $\sqrt{d}$ as $d = 128$, this confirms the theory, but it does not impair the sublinear scaling in $n$ of our algorithm: we could simply scale down $\epsilon$ by a factor of $\sqrt{d}$ to absorb this blowup and increase the runtime by a factor of $\sqrt{d}$.

- The ratio $\frac{d}{\text{srank}(V)}$ is a small constant, recall that $\text{srank}(V)$ can be as small as 1, causing the ratio to be $d$, our experiment shows that $\alpha(V)$ is close to a small constant rather than $d$.

- The ratio $\frac{\|A\|_\infty}{\|A\|}$ is a constant smaller than 3, we check this quantity as in proving the approximation guarantee for $\widetilde{D}$, we make use of the fact that $\|A\|_\infty \leq \sqrt{n} \cdot \|A\|$, this again

shows that instead of the worst case $\sqrt{n}$ scaling, this distortion is only by a constant factor, implying the $\lambda\sqrt{n}$ additive error term is more likely $O(\lambda)$ in practice. This greatly enlarges the range of choice for $\lambda$ to achieve better speedup.

# F    BIT COMPLEXITY OF OUR ALGORITHM

In this section, we give a preliminary analysis on the bit complexity of our algorithm, in particular the bit complexity of matrix inversion operation. We will make use of the following standard algorithm for backward stable matrix inversion.

**Lemma F.1** (Higham (2002); Harvey & van der Hoeven (2021))**.** *Let $A \in \mathbb{R}^{s \times s}$ be nonsingular, there exists an algorithm that computes $B^{-1}$ such that*

$$\|B - A\| \leq \delta \cdot s^c \cdot \kappa(A)^{C \log s} \cdot \|A^{-1}\|,$$

*for absolute constant $c, C > 0$ with bit complexity $O(s^3 \cdot M(b))$ where $b = O(\log(\kappa(A)) + \log(1/\delta))$ and $M(b) = O(b \log b)$.*

We note that the backward stable error guarantee is exactly what has been analyzed in Gu et al. (2024):

**Lemma F.2** (Lemma G.3 in Gu et al. (2024))**.** *Let $A, B \in \mathbb{R}^{s \times s}$ be matrices such that $\|A - B\| \leq \delta$, then*

$$|\tau_i(A) - \tau_i(B)| \leq \delta \cdot \kappa^{2.5}(A).$$

This means that by setting $\delta = 1/\operatorname{poly}(\kappa(A))$, we can approximate the leverage scores well, and the number of bits $b = O(\log(\kappa(A)))$. Note that we apply matrix inversions for two type of matrices:

- $S^\top E S + \lambda I$, where $S$ is the ridge leverage score sampling matrix for $E$;
- $V^\top V$, where $V$ is the value matrix.

In the latter case, we only need to pay the $O(\log(\kappa(V)))$ factor, which in practice, is very small: in our experiments, we see that on average, the log of the condition number is smaller than 4 and the largest log of the condition number is smaller than 20. The interesting part is the former case.

To analyze $\kappa(S^\top E S + \lambda I)$, we upper bound the spectral norm and lower bound the smallest eigenvalue. First, observe that $S^\top E S \succeq 0$, so trivially we have $S^\top E S + \lambda I \succeq \lambda I$, thus the smallest eigenvalue is at least $\lambda$. To bound $\|S^\top E S + \lambda I\|$, we note that

$$\|S^\top E S + \lambda I\| \leq \|S^\top E S\| + \lambda$$
$$\leq \|S\|^2 \cdot \|E\| + \lambda,$$

we bound the two spectral norms respectively. For $\|S\|^2$, we bound it probabilistically: let $c_i = \begin{cases} 1, & \text{if } i \text{ is sampled with probability } p_i \\ 0, & \text{otherwise} \end{cases}$, and consider the matrix $SS^\top$, note that by definition, $SS^\top$ is a diagonal matrix with

$$(SS^\top)_{i,i} = \frac{c_i}{p_i},$$

note that as $c_i$ is a Bernoulli random variable with probability $p_i$, we have $\mathbb{E}[c_i] = 1$ hence $\mathbb{E}\left[\frac{c_i}{p_i}\right] = 1$, and

$$\mathbb{E}[\|S\|^2] = \mathbb{E}\left[\max_{i \in [n]} \frac{c_i}{p_i}\right]$$
$$\leq \mathbb{E}\left[\sum_{i=1}^{n} \frac{c_i}{p_i}\right]$$
$$= \sum_{i=1}^{n} \mathbb{E}\left[\frac{c_i}{p_i}\right]$$

$$= n,$$

hence by Markov's inequality, with constant probability (say 0.99), we have that $\|S\|^2 \leq O(n)$. Condition on this event, we analyze $\|E\|$: let $R = \max\{\max_i \|q_i\|_2^2, \max_i \|k_i\|_2^2\}$, then

$$\|E\| \leq \operatorname{tr}[E]$$
$$= \sum_{i=1}^{n} \exp(\|q_i\|_2^2/\sqrt{d}) + \exp(\|k_i\|_2^2/\sqrt{d})$$
$$\leq 2n \exp(R/\sqrt{d}),$$

combining the above, we obtain a final (probabilistic) upper bound on the condition number of $S^\top E S + \lambda I$:

$$\kappa(S^\top E S + \lambda I) \leq \frac{\|S\|^2 \cdot \|E\| + \lambda}{\lambda}$$
$$\leq 1 + \frac{Cn^2 \exp(R/\sqrt{d})}{\lambda},$$

this gives the final bound on $\log(\kappa(S^\top E S + \lambda I))$:

$$\log(\kappa(S^\top E S + \lambda I)) \leq R/\sqrt{d} + \log(n/\lambda).$$

In practice, the data-dependent parameter $R/\sqrt{d}$ is small: for both OLMo2-1B and OLMo2-7B models, these values are 20.1147 and 21.2227 respectively. Hence, the final bit complexity is $\widetilde{O}(s^3(d^{-0.5}R + \log(\kappa(V)) + \log(n/\lambda)))$.

When performing leverage score sampling over $V$, we need to compute the inverse $(V^\top V)^{-1}$, thus it is mandatory to obtain an upper bound on the condition number of $V$. To compute such an upper bound, we note that the algorithm computes a leverage score sampling matrix $S$ with $O(\epsilon^{-2}d\log d)$ rows, and the matrix $SV \in \mathbb{R}^{\epsilon^{-2}d\log d \times d}$. Computing the condition number and spectral norm of $SV$ can be done classically, in $\operatorname{poly}(d)$ time.

To establish a relation between the conditioning of $V$ and $SV$, observe that $S$ provides a subspace embedding property: $(1 - \epsilon)V^\top V \preceq V^\top S^\top S V \preceq (1 + \epsilon)V^\top V$, this implies that $\kappa(SV) \leq \sqrt{\frac{1+\epsilon}{1-\epsilon}} \cdot \kappa(V) \leq (1 + O(\epsilon)) \cdot \kappa(V)$. This ensures that the bit complexity $b$ depends only on $O(\log(\kappa(V))$.

Finally, to compute the spectral norms and condition numbers required by the algorithms, we could use the algorithms in Musco et al. (2018); Shah (2025); Sobczyk (2025).

## LLM USAGE DISCLOSURE

LLMs were used only to polish language, such as grammar and wording. These models did not contribute to idea creation or writing, and the authors take full responsibility for this paper's content.

