# OpenReview forum: "Sublinear Time Quantum Algorithm for Attention Approximation"
_ICLR.cc/2026/Conference — ICLR 2026 Poster_

### Official Review · Reviewer_zcdq · 2025-10-20

**Soundness:** 3
**Presentation:** 3
**Contribution:** 2
**Rating:** 4
**Confidence:** 4

**Summary:**

This paper studies quantum algorithms (in the QRAM model) to approximate attention in transformers. Given query-key-value matrices $Q,K,V$ of sizes $n\times d,d\times n,$ and $n\times d$, respectively, the goal is to approximate the attention matrix $A=softmax(QK^\top)V$. This is achieved by constructing a data structure that, given any index $i\in[n]$, it returns $\widetilde r_i\in\mathbb{R}^{d}$, which approximates the  $i$-th row of $A$. The total complexity to construct the data structure  depends on $n,d$ and other parameters such as the accuracy, statistical dimension, and row-distortion, and Frobenius norms (I expand further below). If the additional parameters besides $n,d$ are treated as constants, then the proposed approach provides quadratic speed-up with respect to the large dimension $n$, against known algorithms for the problem at hand. Importantly, the complexity guarantees are also accompanied with approximation bounds.

**Strengths:**

1) The paper is well-written and concise, which makes it enjoyable to read.
2) It targets one of the currently most popular computational problems in deep learning, the "quadratic curse" of attention.
3) It provides a novel approach to the problem, combining (non-trivially) techniques from random Linear Algebra and quantum computing.
4) It achieves a complexity that scales as $\sqrt{n}$, with respect to the large dimension $n$. This is impressive, even if it only holds for certain parameter regimes. To my knowledge, classical algorithms require $\Omega(n^2)$ time to achieve sharp element-wise approximations (ref [2] below), or $\Omega(n)$ for less strict approximations.
5) The related work discussion is thorough.
6) The mathematical analysis is very rigorous. I did not read all the details in the proofs, but I was not able to "break" any of them, they seem correct, and well-written.

### References
- [1] Demmel, James, Ioana Dumitriu, and Olga Holtz. "Fast linear algebra is stable." Numerische Mathematik 108.1 (2007): 59-91.
- [2] Alman, Josh, and Zhao Song. "Fast attention requires bounded entries." Advances in Neural Information Processing Systems 36 (2023): 63117-63135.

**Weaknesses:**

I have only two "major concerns" to raise at this stage. Below in "Questions" I provide specific questions that would help me understand the details better and clarify these points.
1) **Classical model of computation**: From what I understand, there are subroutines in the main algorithm that rely on classical computations. For example, line 274 assumes a subroutine to compute the pseudoinverse in $O(n^\omega)$. How? From what I know, finite precision algorithms can only return approximate solutions (see e.g. ref [1] below). Things might be easier in "exact arithmetic", but I am not sure that infinite-precision is compatible with QRAM.
2) **Approximation/complexity trade-off**: I am a bit sceptical about the complexity / approximation trade-off. In Theorem 3.1, line 362, there is a $\sqrt{n}$ term, and two more "hidden" in the Frobenius norms. The former can be absorbed by setting e.g. $\lambda=1/\sqrt{n}$, but it is a bit unclear how this affects complexity. Now, if we were to upper bound the $||\cdot||_F$-norms with $||\cdot||_2$-norms, which is commonly the desired type of bound, they would intruduce another $\sqrt{n}$ factor. This factor would have to be absorbed inside $\epsilon$, e.g., by setting $\epsilon'=\epsilon/\sqrt{n}$. But they this would introduce an additional $\sqrt{n}$ factor in the complexity of line 365, and therefore it would no longer be sublinear in $n$.

**Minor concerns:**
I have the following two minor comments (but they did not influence my recommendation).

- The QRAM model is mostly of theoretical interest (at least at the time of this writing). This might be a limitation for practical implementations in the future.
- The authors recognize in line 377 that the reported approximation guarantee is for a "symmetrized version" rather than the classic norm-wise approximation. I think this is fine, I am more concerned with the Frobenius-versus-spectral norm topic, as I mentioned above. But it certainly speaks in favor of the authors that they explicitly mention this topic.

**Questions:**

My current assessment is slightly leaning towards reject, due to the two main concerns that I raised above. At this stage it is not clear to me if the final, end-to-end complexity achieves the reported sublinear time, or, if it does, what are the corresponding parameter regimes. My recommendation is not final. I will take into consideration the authors responses as well as the comments from the other reviewers.
Here I mention some questions that I would like answered to help me clarify my understanding of the paper and provide the additional evidence for my final assessment.
### Questions
1) Regarding concern 1): What is the "classical" model of computation followed here? Is it "compatible" with QRAM? Could you provide references/discussion on the precise complexity / approximation guarantees of the assumed classical subroutines?
2) Regarding concern 2): Could you provide a small paragraph discussing further how the choice of the different parameters affects the total complexity? If someone wants spectral-norm bounds, how can they be achieved?
3) Can we replace QRAM with something simpler (e.g., QROM)? Which parts are currently the "bottleneck"?
4) Are there any quantum / classical lower bounds for Frobenius-norm type of approximations, e.g., in similar spirit to [2] below. I do not expect the authors to prove lower bounds at this stage, but a relevant discussion would be helpful.
5) If we were to use the proposed algorithms to  approximate the entire attention matrix, what would be the complexity and how does it compare with existing attention algorithms? I think that the $\Omega(n^2)$ lower bounds of [2] leave quite some room for improvements. E.g., if the final complexity of the proposed algorithm is $O(n^{1.5})$ to achieve the same (or similar) bound as [2], then this would already be a nice improvement, and would significantly strengthen the presentation. Could you provide some insights?
6) Could the authors comment on how to choose the $\lambda$ parameter?

### Additional Feedback
Here I provide additional feedback with the aim to improve the paper. These points are here to help, and not necessarily part of the decision assessment.

- The main result, Theorem 3.1, is in page 7. It would be nice to either move it earlier, e.g., in the introduction, or at least a more explicit statement of the main result in the introduction.
- A table with the main result compared to existing algorithms could be helpful. E.g., to compare complexity, approximation guarantees (if any), the model of computation, or other properties that the authors consider important (again, this is not a request for the rebuttal, just potentially nice-to-have)
- Some paragraphs are a bit long, e.g., the first paragraph of Section 3.3.
- Using colored references can be helpful, and I think it is allowed by ICLR template.
- In line 206, $O(s^2)\cdot \mathcal{T}_K+s^\omega$ should be $O(s^2\cdot \mathcal{T}_K+s^\omega)$. The hidden constants in fast matrix product are quite large. There might be other places in the paper where this applies.
- When mentioning algorithms/subroutines with $O(n^\omega)$  complexity, a reference or proof should be given. I know that some theory papers tend to take them for granted, but often they are highly non-trivial to prove, or even to find the corresponding bibliography.
- Between lines 368-369: The sentence "...achieving a quadratic speedup over any classical algorithm" should probably be "...achieving a quadratic speedup with respect to $n$ over any classical algorithm that we know of".
- In line 372, it is mentioned "when $a=o(n)$". Based on Definition C.2 in the Appendix, I think $a(V)$ is always at most $d$. Take the SVD of $V=U\Sigma W^\top$, and let $v_i$ be the $i$-th row of $V$. It holds that
$||v_i||_{2}^{2}=||e_i^{\top}U\Sigma W^{\top}||_2^2\leq  ||e_i^{\top} U||_2^2||\Sigma W^\top||_2^2=\tau_i||V||_2^2.$ Replacing this in Definition C.2 gives
$\frac{d}{||V||_F^2}\cdot \max_i\frac{||v_i||_2^2}{\tau_i} \leq d$.

**Details Of Ethics Concerns:**

No ethics concerns.

---

> ### Author Response · Authors · 2025-11-15
>
> We appreciate your very detailed review and insightful comments, in particular the upper bound on $\alpha(V)$. We are happy to report that we in fact remove the symmetrization error guarantee, and obtain an actual Frobenius norm error guarantee in the form of $\\|\widetilde D^{-1} \widetilde A \widetilde V - {\rm Att}(Q, K, V)\\|_F$, we refer to the global response for a more detailed discussion.
>
> * Regarding classical model of computation and fast matrix multiplication: Thank you for raising this point. Indeed, fast matrix multiplication would incur error when implemented with finite precision, and to recover the exact inverses and pseudoinverses, infinite precision is needed. There are many works analyzing the bit complexity of algorithms that make use of fast matrix multiplication-based operations, see [4, 5, 6]. In the QRAM model, [7] also uses fast matrix multiplication-based inversion without further discussions. While we believe it’s possible to analyze the bit complexity of these operations in our algorithm in the QRAM model, it is currently beyond the scope of our work. Fortunately, we note that in our work, we only apply fast matrix multiplication-based inversion on small matrices of size $d\times d$ or $s\times s$, hence, if we instead use the cubic time matrix multiplication algorithm, it would only slightly worsen the runtime dependence on the lower order terms, specifically from $s^\omega+d^\omega$ to $s^3+d^3$. In our revised draft, we have adopted the cubic time complexity for these operations. Note that this does not affect our overall sublinear runtime, as these polynomial factors are on the lower order terms. Finally, for references on the $n^\omega$ complexity of matrix inversion, see [9], chapter 16, for a long list of matrix operations that can be done in $n^\omega$ time (in exact arithmetic).
>
> * Regarding approximation and complexity trade-off: Indeed, in our final error bound, there is a $\sqrt{n}$ term and the error is given in terms of Frobenius norm instead of spectral norm. In particular, naively converting $\\|A\\|_F$ to $\\|A\\|$ would incur an additional $\sqrt{n}$ factor. Theoretically, these factors are tricky to remove: the $\sqrt{n}$ factor for $\lambda$ comes from converting the $\lambda$-spectral norm error bound to matrix infinity norm, and the Frobenius norm is due to the use of quantum leverage score sampling solely on the matrix $V$. To instead obtain a spectral norm bound, one could use the quantum leverage score sampling on the column span of $\begin{bmatrix} D^{-1}A & V\end{bmatrix}$, however this would significantly inflate the required sample complexity from $\widetilde O(\epsilon^{-2}\alpha)$ to $\widetilde O(\epsilon^{-2} {\rm srank}(D^{-1}A))$, which, unlike statistical dimension, is not a tunable parameter and could potentially be as large as $n$. On the other hand, we note that these estimates are often pessimistic, on actual attention modules, these bounds could be much smaller. We empirically verify this conjecture, specifically, we compare the spectral vs Frobenius norm of $A$ and $V$ matrices, and spectral vs infinity norm of $A$. For the former, we observe that for $A$, these two norms often only differ by a factor of 2, much smaller than the predicted $\sqrt{n}$ factor. For $V$ however, we could see a quite clear $\sqrt{d}$ factor gap. Nevertheless, we could set $\epsilon’=\epsilon/\sqrt{d}$ to account for this gap while maintaining the sublinear complexity. For the spectral vs infinity norm, we again observe a factor of at most 3, implying that instead of the pessimistic $\lambda\sqrt{n}$ predicted by theory, the additive error term is more likely to be $O(\lambda)$ in practice. For more detailed discussion, we refer you to the global response section. Thus, to answer your question regarding how to achieve a spectral norm error guarantee, a practical answer would be scaling down $\epsilon$ by a constant factor, rather than $\sqrt{n}$.
>
> * Regarding QRAM: We agree that the QRAM model is considered strong. On the other hand, if instead of time complexity, one counts the query complexity to the rows of $Q, K, V$, it is independent of the QRAM model. In particular, we show that our algorithm makes $\widetilde O(\epsilon^{-1} n^{0.5}s^{0.5})$ row queries to $Q, K$ and $\widetilde O(\epsilon^{-1}n^{0.5}\alpha^{0.5})$ row queries to $V$, and we have incorporated the revision into the draft.
>
> * Regarding lower bound for Frobenius norm type approximation: We are not aware of any Frobenius norm lower bound similar to Alman and Song. One possible way to derive a lower bound is to follow the quantum query lower bound of [7] that gives such a guarantee for computing a spectral approximation, which would yield a lower bound for our kernel Nystrom step. In general, it’s much harder to establish a Frobenius norm lower bound for the attention approximation problem without restricting the class of algorithms. We leave this as a major future direction.

---

> > ### Author Response · Authors · 2025-11-15
> >
> > * Regarding turning our algorithm into outputting the full attention matrix: If we are to output the entire attention matrix, then the overall runtime is $\widetilde O(\epsilon^{-1}n^{0.5}(s^{2.5}+s^{1.5}d+\alpha^{0.5}d))+\widetilde O(n(s^2+sd)))$, i.e., the dependence on $n$ is $\epsilon^{-1}n^{0.5}$ and $n$, not $n^{1.5}$, as the $\epsilon^{-1}n^{0.5}$ factor only comes up during the preprocessing phase, and each row query can be done in $\widetilde O(s^2+sd)$ time. If one wants to convert the Frobenius norm bound (or even the stronger spectral norm bound) to the entrywise infinity norm bound of Alman and Song, we have $\\|{\rm Att}(Q, K, V)\\|\leq \\|{\rm Att}(Q, K, V)\\|_F\leq \sqrt{nd}\cdot \\|{\rm Att}(Q, K, V)\\|\_{\max}$, which means that we have to scale down $\epsilon$ by a factor of at least $\sqrt{nd}$. Moreover, the approximation guarantee of Alman and Song is purely additive, i.e., they measure $\\|\widetilde D^{-1}\widetilde A\widetilde V-{\rm Att}(Q, K, V)\\|\_{\max}\leq \epsilon$, while for our approximation guarantee and other subquadratic attention approximation schemes [8, 9] obtain errors relative to the spectral or Frobenius norms of the modules. Converting these relative error terms to absolute additive error would require scaling down $\epsilon$ by norms of these matrices, which could be as large as ${\rm poly}(n)$, completely invalidating the efficiency gain. In fact, the bounded entry assumption in Alman and Song is one way to control the magnitude of these norms: they assume all entries of $Q, K, V$ are $o(\sqrt{\log n})$ for almost linear time algorithms, this would imply the matrix $A$ has entries of magnitude $o(n)$, implying an $o(n^2)$ upper bound on the Frobenius and spectral norms of $A$. In the “lower bound” regime where the entries are $\Theta(\sqrt{\log n})$, we note that $\\|A\\|_F = O(n^2)$, if we assume $\\|D^{-1}\\|=0.1\frac{1}{\epsilon \\|A\\|}$ by properly choosing $\epsilon$, so that $\beta \cdot \\|D^{-1}\\|=O(1)$, then we would need to set $\epsilon=O(1/n^2)$, which means the runtime can be as large as $O(n^{2.5})$. To achieve a subquadratic scaling, we would require $\\|A\\|_F\leq o(n^{1.5})$. We would like to note that in our experiment, the Frobenius norm of $\\|A\\|_F$ is much smaller: for $n=4096$, the Frobenius norm is only roughly 10000, which is much smaller than the worst case scaling predicted by the theory. We would also like to remark that the additive error in the matrix max norm is a very strong guarantee compared to the more common relative spectral norm error or our relative Frobenius norm, thus it would be interesting to investigate a lower bound on this type of more practical error guarantee.
> >
> > * Regarding the choice of $\lambda$: in practice, the process is usually to first determine the sample size $s=k \log k$ for some integer $k$, then determining the value of $\lambda$ as $\lambda=\frac{1}{k}\sum_{i=k+1}^{2n} \sigma_i(E)$, see [3], section 5 for a discussion.
> >
> > * Regarding additional feedback: Thank you for raising these points to improve the presentation of our paper, we have addressed them accordingly. We have fixed the upper bound on $\alpha(V)$ and prove that $\alpha(V)\leq \frac{d}{{\rm srank}(V)}$, see the global response for more details.
> >
> >
> > References
> >
> > [3] Recursive sampling for the Nystrom method. Musco and Musco. NIPS 2017.
> >
> > [4] The bit complexity of efficient continuous optimization. Ghadiri, Peng and Vempala. FOCS 2023.
> >
> > [5] Improving the Bit Complexity of Communication for Distributed Convex Optimization. Ghadiri, Lee, Padmanabhan, Swartworth, Woodruff and Ye. STOC 2024.
> >
> > [6] The Bit Complexity of Dynamic Algebraic Formulas and their Determinants. Anand, van den Brand, Ghadiri and Zhang. ICALP 2024.
> >
> > [7] Quantum speedups for linear programming via interior point methods. Apers and Gribling. QIP 2024.
> >
> > [9] Algebraic complexity theory. Burgisser, Clausen and Shokrollahi. 1997.

---

> ### Comment · Reviewer_zcdq · 2025-11-17
> **Follow-up on classical model of computation**
>
> Let us expand a bit more on the classical model of computation. From what I understand QRAM can read classical **bits** from the memory (potentially in superposition). It can also write back classical **bits**. I am concerned if we have a hybrid quantum-classical algorithm can do both of the following simultaneously:
> - do classical operations {+,-,*,/} in exact arithmetic (real numbers) in constant time, (this seems to be assumed for the pseudo-inverse computation around line ~270, regardless whether it uses an $n^\omega$ or $n^3$ type algorithm).
> - treat the real numbers as bitstrings, and store them in QRAM
>
> I may be a bit too picky on this, and/or my concern might also be unjustified, but I would really like to understand what happens in this case. At this point, a bit complexity analysis of the pseudoinverse seems necessary, or a convincing explanation why this is not really an issue.

---

> > ### Author Response · Authors · 2025-11-18
> >
> > Thank you for the followup question. We would like to clarify that our model treats the real numbers as finite-precision $b$-bit words, and arithmetic operations would require $M(b)$ time where $M(b)=O(b^2)$ or $O(b\log b\log \log b)$ if one uses fast multiplication, and the inverses/pseudoinverses would take $O(s^3 M(b))$ time where $b=\log(\kappa(M))+\log(1/\delta)$. If for an $s\times s$ matrix $M$ and its target inverse $M^{-1}$, we compute $\widetilde M^{-1}$ with $\\|\widetilde M-M\\|\leq \delta$, this can be done via a backward stable algorithm to solve the linear system [10, 11]. To ensure the ridge leverage score and leverage score sampling gives the correct distribution, it is enough to choose $\delta=1/{\rm poly}(\kappa(M))$, see Appendix G of [12] for a detailed analysis. Note that in two places we use inversion: one for $S^\top E S+\lambda I$, where $S$ is the ridge leverage score matrix, the other for the leverage score for the matrix $V$. In the latter case, the condition number is just $\kappa(V^\top V)=\kappa^2(V)$, and we empirically verify that $\log(\kappa(V))$ is typically very small ($<20$). For the matrix $S^\top ES+\lambda I$, we bound its condition number as follows: as $S^\top ES$ is PSD, we know that its minimum eigenvalue is at least $\lambda$, and to obtain an upper bound, we see that $\\|S^\top ES+\lambda I\\|\leq \\|S\\|^2\cdot \\|E\\|+\lambda$, so the condition number is upper bounded by $1+\frac{\\|S\\|^2\cdot \\|E\\|}{\lambda}$. While the spectral norm of $E$ is dependent on the input $Q, K$, the spectral norm of the ridge leverage score sampling matrix is a bit tricky. Note that $\mathbb{E}[SS^\top]=I$ and $\mathbb{E}[\\|S\\|^2]\leq n$, by Markov’s inequality, with constant probability, $\\|S\\|^2\leq O(n)$. Conditioning on this event, we see that the condition number of $S^\top ES+\lambda I$ is upper bounded by $\frac{Cn\\|E\\|}{\lambda}$ for some constant $C>0$. To give a simple upper bound on $\\|E\\|$, note that $\\|E\\|\leq {\rm Tr}[E]=\sum_i \exp( \\|q_i\\|\_2^2/\sqrt{d})+\exp(\\|k_i\\|\_2^2/\sqrt{d})$, so if we assume a radius on the queries and keys $\max\\{\max_{i} \\|q_i\\|_2^2, \\|k_i\\|_2^2 \\}\leq R$, then we see that the spectral norm can be upper bounded $2n \exp(R/\sqrt{d})$. Hence, an upper bound on the condition number becomes $\frac{O(n)\cdot \exp(R/\sqrt{d})}{\lambda}$, and the overall dependence on $b$ becomes $R/\sqrt{d}+\log (n/\lambda)$ by taking log. Hence, we can conclude the overall runtime as $\widetilde O(s^3(R/\sqrt{d}+\log(n/\lambda)+\log\kappa(V)))$. We empirically verify that $R/\sqrt{d}$ is roughly 20 for both 1B and 7B, hence this data-dependent blowup is very small. We have added a section in the appendix (Appendix F) to discuss the bit complexity.
> >
> > [10] Complexity of computations with matrices and polynomials. Pan. SIAM Review, 1992.
> >
> > [11] Accuracy and Stability of Numerical Algorithms. Higham. SIAM, 2002.
> >
> > [12] Low rank matrix completion via robust alternating minimization in nearly linear time. Gu, Song, Yin and Zhang. ICLR 2024.

---

> ### Comment · Reviewer_zcdq · 2025-11-18
>
> Ok thank you for the response. I think it is going on the right direction but I am not sure we are fully there yet. I especially appreciate that the dependence on the radius $R$ showed up, it makes the arguments more convincing.
> It looks like you followed a "floating-point" type of bit complexity analysis. I have these follow-up questions:
> 1) From what I know, floating point analysis of matrix inverse should give a "backward-approximation" that depends on the floating point accuracy, the matrix dimension, the condition number, and the matrix norm. See e.g. Definition 2.7 of [2], which builds on the results of [3] (I cited the Arxiv versions here, if you decide to cite this works, please use the published versions). I am not sure if the claim that $||M-\tilde M||\leq \delta$ is true. I would expect something like $||M-\tilde M||\leq poly(n,\kappa(M)) \cdot \delta \cdot ||M||$.
> 2) If you need to scale $\epsilon$ by $\kappa(M)$ or $||M||$, how do you compute $\kappa(M)$ and/or $||M||$, respectively? If the matrix has size $d\times d$, then I believe that you can do it in $poly(d)$ which does not affect the "sublinearity" in $n$. But if you need to compute the norm of $V$, for example, then I am not sure if you can do it in $o(n)$. Is there a sublinear-time quantum algorithm to compute the spectral norm?
>
> I also have the following two remarks that could be helpful:
> - You can potentially control the condition number of any matrix to make it polynomial by adding a random perturbation, in the spirit of "pseudospectral shattering" of [1] (potentially other works have refined the analysis for sparse and/or symmetric matrices)
> - **Nitpick**: the best known bound for integer multiplication is currently $O(b\log(b))$, see [1]
> - Do not feel obliged to update the paper every time we discuss about these issues. I am more than happy if you add them in the final version, as long as we reach an agreement during the discussion.
>
> #### Refs
> - [1] Harvey, David, and Joris Van Der Hoeven. "Integer multiplication in time O(nlog\,n)." Annals of Mathematics 193.2 (2021): 563-617.
> - [2] Banks, Jess, et al. "Pseudospectral Shattering, the Sign Function, and Diagonalization in Nearly Matrix Multiplication Time." arXiv preprint arXiv:1912.08805 (2019). https://arxiv.org/abs/1912.08805
> - [3] Demmel, James, Ioana Dumitriu, and Olga Holtz. "Fast linear algebra is stable." arXiv preprint math/0612264 (2006).

---

> > ### Author Response · Authors · 2025-11-18
> >
> > Thank you for pointing out useful references for bounding the backward error. Using [13], we see that for matrix $A\in \mathbb{R}^{s\times s}$, we can compute a matrix $B^{-1}$ such that $\\|B^{-1}-A^{-1}\\| \leq \delta\cdot s^{c}\kappa(A)^{C\log n} \\|A^{-1}\\|$ where $c, C>0$ are absolute constants. As you have noted, when working with $S^\top ES+\lambda I$, this is a matrix of size $s\times s$, so we can compute these quantities in $o(n)$ time.
> >
> > For $V$, there are two routes: (1) using quantum algorithms to approximate the spectral norm and condition number, or (2) instead computing the spectral norm and condition number for the subsampled matrix. For the first route, we note that the error of approximating the inverse of $V$ is $\delta\cdot d^c \kappa(V)^{2C\log n} \sigma_{\min}(V)^{-2}$, so we need to at least upper bound $\kappa(V)$ and $\sigma_{\min}(V)^{-2}$ in $o(n)$ time. To estimate the spectral norm of $V$, we could make use of [14], where Theorem 1 shows that one could estimate $\\|V\\|$ up to $\epsilon \\|V\\|\_F$ Frobenius norm error in time $\widetilde O(\frac{\log(1/\epsilon)}{\epsilon} \frac{\\|V\\|\_F}{\\|V\\|})$, as $\\|V\\|\_F\leq \sqrt{d} \\|V\\|$, this runtime reduces to $\widetilde O(\frac{\log(1/\epsilon)}{\epsilon}d^{0.5})$ and we could scale down $\epsilon$ by a factor of $\sqrt{d}$ to ensure at least constant relative error. To approximate the smallest singular value, we use the procedure described in [14] that finds a threshold $\theta$ such that $\sum_{i: \sigma_i\leq \theta} \sigma_i^2/\\|V\\|\_F^2\geq\sigma_{\min}(V)^2/\\|V\\|\_F^2$ with additive error $\epsilon$, in $\widetilde O(\frac{\\|V\\|\_F}{\sigma\_{\min}(V)}\frac{\log(1/\epsilon)}{\epsilon})$ time, note that $\\|V\\|\_F/\sigma\_{\min}(V)\leq \sqrt{d} \cdot \kappa(V)$, so the dependence on condition number is linear, which could be undesirable. On the other hand, linear or polynomial dependence on $\kappa(V)$ is common in quantum algorithms [14, 15].
> >
> > Alternatively, one could observe that when running the recursive leverage score sampling for $V$, we *never* explicitly need the matrix $(V^\top V)^{-1}$, all we need is $(V^\top S^\top SV)^{-1}$, where $S$ is a sampling matrix with $O(\epsilon^{-2}d\log d)$ rows and it provides the subspace embedding property: $(1-\epsilon) V^\top V \preceq V^\top S^\top SV \preceq (1+\epsilon) V^\top V$, so it’s easy to see that $\kappa(SV)\leq \sqrt{\frac{1+\epsilon}{1-\epsilon}} \kappa(V)$. Thus, we could express the bit complexity on the $V$ part in terms of $\kappa(SV)$, which could be computed in ${\rm poly}(d)$ time, and we maintain an $O(\log(\kappa(V)))$ dependence on $b$.
> >
> > We will make sure to properly cite the references you have pointed out. Thank you again for the discussion.
> >
> > [13] Pseudospectral Shattering, the Sign Function, and Diagonalization in Nearly Matrix Multiplication Time. Banks, Garza-Vargas, Kulkarni and Srivastava. FOCS 2020.
> >
> > [14] Quantum gradient descent for linear systems and least squares. Kerenidis and Prakash. Physical Reviews A, 2020.
> >
> > [15] Quantum algorithm for linear systems of equations. Harrow, Hassidim and Lloyd. PRL 2009.

---

> > > ### Comment · Reviewer_zcdq · 2025-11-19
> > >
> > > Ok great - the subspace embedding argument for $V$ is quite convincing and very clean, I admit I did not think of that (just please make it rigorous in the final version). My concern about the "classical subroutines" is fully covered, thank you for the nice discussion. Some final side notes:
> > > - [13] does not actually give an algorithm for inverse, they use the result of https://arxiv.org/pdf/math/0612264 (see Eq. (9))
> > > - These references might be helpful for the bit complexity of actually computing the spectral norm / condition number (for the sake of "scrutiny"):
> > >   - https://arxiv.org/abs/1708.07788
> > >   - https://arxiv.org/abs/2408.09880
> > >   - https://arxiv.org/abs/2410.21550
> > >
> > > I need to look in more detail to the comments of the authors regarding the other questions. They seem quite convincing with a quick look but I did not have time to look at it rigorously yet. I will make a final recommendation in the next few days, thanks again!

---

> > > > ### Comment · Reviewer_zcdq · 2025-11-19
> > > > **Update on decision after rebuttal**
> > > >
> > > > The authors addressed all my concerns, especially the major one regarding the classical model of computation. I think the paper is now very robust and a very nice contribution. I update my score from 4 to 8. I am confident with my evaluation and I will insist on acceptance.
> > > >
> > > > I trust that the authors will carefully address all the discussions in the final version. At the same time, I want to note that the paper went through a "major revision" throughout the rebuttal so far. I did not appreciate this - evidently, the paper was not ready for submission. In the future I might decide to recommend rejection of other papers for this reason. But for the sake of having high quality papers in ICLR 2026, I strongly recommend acceptance.

---

> > > > > ### Author Response · Authors · 2025-11-20
> > > > >
> > > > > Thank you very much for the very engaging discussion that helps significantly improve the clarity, rigor and technical depth of our draft. We will make sure to incorporate all the changes in the final version, and include all the necessary references. Thank you again for your help!

---

### Official Review · Reviewer_eeim · 2025-10-21

**Soundness:** 3
**Presentation:** 3
**Contribution:** 2
**Rating:** 6
**Confidence:** 3

**Summary:**

This paper proposes a quantum data structure for approximating rows of the attention matrix in sublinear time with respect to the sequence length n. The method combines quantum Nyström approximation, multivariate mean estimation, and leverage score sampling to approximate the components of the attention mechanism. This is the first quantum algorithm to achieve sublinear dependence on n in the row-query model without structural assumptions.

**Strengths:**

1. The work is the first to achieve sublinear-in-n row queries for attention approximation using quantum methods.
2. The approach makes no  structural assumptions making it widely applicable.

**Weaknesses:**

1. Parameter dependence: The runtime depends on s  and α , which may be large in practice, limiting practical speedups.
2. Norm of D−1 assumption: The guarantee requires ∥D−1∥<(ϵ∥E∥+λn)−1, which may not hold in all settings.

**Questions:**

1. Can you give numberical experiments to show the time cost, errors and the assumptions on parameters.
2. How does the statistical dimension s behave in practice for typical transformer inputs, and does it remain small enough to yield meaningful speedups?
3. Is the row distortion parameter α bounded in real-world value matrices, and are there cases where it becomes large enough to negate the sublinear advantage?

---

> ### Author Response · Authors · 2025-11-15
>
> We appreciate your valuable comments. We have significantly improved the results obtained in the paper and we refer you to the global response for an overview. Below, we address your specific concerns and questions.
>
>
> * Regarding the parameter $\alpha$: Thank you for raising this point. We have proved that the parameter $\alpha(V)\leq \frac{d}{{\rm srank}(V)}$ which is at most $d$, and empirically verify that it is usually a small constant smaller than 3. We refer to a more detailed discussion in the global response.
>
> * Regarding the statistical dimension $s$, we note that a typical practical pipeline is to choose $s$ first, then determine $\lambda$ based on that, see [3], section 5 for a discussion.
>
> * Regarding the $\\|D^{-1}\\|$ assumption: See our empirical verifications in the global response, we have verified that the assumption would hold as long as $\epsilon\leq 0.18$ for practical models and datasets.
>
> * Regarding numerical experiments: While our algorithm and its guarantees rely on QRAM, we perform experiments to test the assumptions on the parameters. We test the $\\|D^{-1}\\|$ assumption, Frobenius norm vs spectral norm of $A$ and $V$, and the upper bound on the row distortion parameter $\alpha(V)$. See the global response for more discussions.
>
> References
>
> [3] Recursive sampling for the Nystrom method. Musco and Musco. NIPS 2017.

---

> > ### Comment · Reviewer_eeim · 2025-11-25
> >
> > Thank you for your reply. As shown in Table 3, we observe that the error for each attention head is significant (about 0.2 for the max error). In a real transformer network, these errors can accumulate through the layers, leading to a larger error in the final output and thus limiting the model's practical applicability.

---

> > > ### Author Response · Authors · 2025-11-25
> > >
> > > Thanks for the response, we would like to clarify that the $\epsilon_{\max}$ in Table 3 doesn't mean that the error per head is $\epsilon_{\max}$, it is defined as $\max_{\epsilon} \\|D^{-1}\\|\leq \frac{1}{\epsilon \\|A\\|}$, i.e., the maximum $\epsilon$ needed in order for the condition on $\\|D^{-1}\\|$ to hold. We can choose **any $\epsilon\leq \epsilon_{\max}$** to satisfy this condition. We want to show that $\epsilon_{\max}$ is not too small, as our runtime scales with $\epsilon^{-1}$. In practice, one could choose any $\epsilon\leq \epsilon_{\max}$ to decrease the possible error accumulation across heads and across layers, e.g., $\epsilon=10^{-2}$ or $\epsilon=10^{-3}$. We hope this clarifies your concerns.

---

> > > > ### Comment · Reviewer_eeim · 2025-11-27
> > > >
> > > > The authors addressed my concerns well. I'm maintaining my positive score.

---

### Official Review · Reviewer_ojWR · 2025-10-26

**Soundness:** 3
**Presentation:** 3
**Contribution:** 2
**Rating:** 8
**Confidence:** 2

**Summary:**

The goal of the paper is to study approximation algorithms for self-attention computation in the transformer architecture. The inputs to self-attention are $Q,K,V \in \mathbb{R}^{n\times d}$ and the goal is to output $Att(Q,K,V) = D^{-1}A V$ where $A= exp(QK^T/\sqrt{d})$ and $D^{-1} = diag(A\mathbb{1})$. Past works for provable attention approximation need at least $\Omega(nd)$ time, which is the input and output size, and the paper focuses on quantum algorithms that can achieve a better runtime. If one insists on outputting the entire $Att(Q,K,V)$ matrix, $\Omega(nd)$ time is inevitable, however this can be avoided by formulating the problem as a data structure problem. In particular the goal is preprocess $Q,K,V$ into a data structure that then allows, for any index $i\in [n]$, the return an approximation to the $i^{th}$ row of $Att(Q,K,V)$. Even then since each row of $Att(Q,K,V)$ is a convex combination of rows of $V$, achieving sublinear in $n$ time is hard.

Their main contribution is a quantum data structure that access the input matrices only using row queries, performs preprocessing in $\widetilde{O}(\epsilon^{-1} n^{0.5} poly(d,s_{\lambda},\alpha)$ time, and answers output row queries in time $\widetilde{O}(s_{\lambda}^2 + s_{\lambda}d)$ (here $s_{\lambda}$ is the statistical dimension of $exp(QK^T/\sqrt{d})$. Their approach uses techniques such as Grover search, Quantum Nystrom approximation, and Quantum multivariate mean estimation.

**Strengths:**

The main strength of the paper is to present a sublinear time algorithm that answers row queries for attention approximation in the quantum model. The techniques are very interesting and conceptually simple.

**Weaknesses:**

Perhaps one minor weakness is that there are few previous works on attention approximation that achieve spectral norm approximation guarantees and it would be to prove such a guarantee here as well.

**Questions:**

The first question is that the authors make a statement that achieving sublinear in $n$ dependence for the row query model seems intractable for classical algorithms since each row of the output is a convex combination of rows of $V$. Is there a formal claim to show this ? If yes then since there are past works on attention approximation that make structural assumptions on the input matrices, is it possible to prove classical sublinear in $n$ guarantees under plausible assumptions ?

---

> ### Author Response · Authors · 2025-11-15
>
> We appreciate your comments and are happy that you find our work interesting and technically interesting and conceptually simple. We have significantly improved the results obtained in the paper and we refer you to the global response for an overview. Below, we address your specific concerns and questions.
>
> * Regarding spectral norm approximation guarantees: We note that the only place in our analysis that incurs a Frobenius norm approximation guarantee is to use leverage score sampling on the value matrix $V$ solely for approximation matrix multiplication. There are two ways to realize a spectral norm approximation (1) use our quantum leverage score sampling algorithm on the column span of $[V; D^{-1}A]$, however, to achieve a desired spectral norm approximation, the number of samples to take needs to be near-linear in the stable rank of $D^{-1}A$ which could be as large as $n$. Alternatively, one could again use ridge leverage score sampling to get a dependence in terms of the statistical dimension of $D^{-1}A$ and extra additive error on $\lambda$, (2) develop a quantum version of the row sampling algorithm similar to [1], this would yield a sample complexity near-linear in the stable rank of $V$ and $D^{-1}A$. However, to compute the sampling probability, one needs the spectral norms and Frobenius norms of $V$ and $D^{-1}A$. To approximate these quantities, [1] uses fast kernel density estimation to approximate the Frobenius norm of $D^{-1}A$, which is unclear how to do so in the quantum setting. Hence, while we believe achieving a spectral norm guarantee is possible, it requires nontrivial ideas such as developing a fast quantum algorithm for kernel density estimation, for which we leave as a future work. Empirically, we verify that the Frobenius and spectral norms of $A$ are only a constant factor off, while for $V$, there exists a $\sqrt{d}$ gap. See the global response for a more detailed discussion.
>
> * Regarding classical algorithms that achieve sublinear time for row queries: [2] shows that for $d=O(\log n)$ and the entries of $Q, K, V$ are $\Theta(\log n)$, then assuming the Strong Exponential Time Hypothesis, there is no algorithm to approximate ${\rm Att}(Q, K, V)$ in $1/{\rm poly}(n)$ additive error in $O(n^{2-o(1)})$ time. Under this setting, one could derive a similar lower bound for the row query model: if there exists a row query algorithm with such guarantee in $O(n^{1-o(1)})$ time, then it would directly imply an $O(n^{2-o(1)})$ time algorithm for the attention approximation problem posed in [2]. On the other hand, we are not aware of any classical algorithm that under plausible assumptions, achieves sublinear time in the row query model.
>
> References
>
> [1] KDEformer: Accelerating Transformers via Kernel Density Estimation. Zandieh, Han, Daliri and Karbasi. ICML 2023.
>
> [2] Fast attention requires bounded entries. Alman and Song. NeurIPS 2023.

---

### Official Review · Reviewer_ppDN · 2025-10-29

**Soundness:** 2
**Presentation:** 3
**Contribution:** 2
**Rating:** 4
**Confidence:** 3

**Summary:**

The paper proposes a quantum data structure for approximating the Transformer attention mechanism under the row query model, where only individual rows of the attention output are queried. The key contribution is a theoretical framework that achieves sublinear preprocessing time complexity of $\tilde{O}(\epsilon^{-1} n^{0.5}(s_\lambda^{2.5} + s_\lambda^{1.5} d + \alpha^{0.5} d))$, providing a quadratic speedup over the best known classical algorithms.
The method embeds the non-symmetric attention matrix $A = \exp(QK^\top / \sqrt{d})$ into a larger symmetric exponential kernel matrix over the combined dataset $(Q, K)$, and applies a combination of quantum Nyström approximation, quantum multivariate mean estimation, and quantum leverage score sampling to approximate the attention normalization factor, kernel matrix, and value multiplication components, respectively.
The resulting data structure allows approximating any attention row in time $\tilde{O}(s_\lambda^2 + s_\lambda d)$, without assumptions on $Q, K, V$. This is, to the authors’ knowledge, the first quantum algorithm achieving sublinear dependence on sequence length $n$ for attention approximation. The authors also provide theoretical guarantees in Frobenius norm for the symmetrized attention matrix $(A + A^\top)/2$, along with detailed parameter dependence on the kernel’s statistical dimension $s_\lambda$ and value distortion factor $\alpha$.

**Strengths:**

S1: High Originality and Theoretical Significance: This work, to the best of my knowledge, is the first to propose a sublinear-time quantum algorithm for approximating the standard Transformer attention mechanism in the row-query setting. Achieving a preprocessing complexity of $\tilde{O}(n^{0.5})$, the method provides a potential quadratic speedup over classical algorithms. This represents a meaningful theoretical advance and offers a new perspective on overcoming the quadratic bottleneck in large-scale attention computation.

S2: Sophisticated Theoretical Framework: The paper demonstrates strong technical depth by systematically combining several advanced quantum tools—Nyström kernel approximation, multivariate mean estimation, and leverage score sampling—into a coherent data structure for attention approximation. The approach of embedding the non-symmetric attention matrix into a symmetric exponential kernel over the joint query–key space is both elegant and conceptually novel. Moreover, the framework is general, requiring no structural assumptions on $Q$, $K$, or $V$, which enhances its theoretical robustness and potential applicability.

**Weaknesses:**

W1: Lack of Empirical Validation: The paper is entirely theoretical and does not provide any numerical simulation or small-scale experiment to illustrate the potential practical impact of the proposed method. While this is acceptable for a theoretical contribution, even a simple empirical demonstration (e.g., simulated quantum runtime scaling or synthetic kernel approximation) would help substantiate the claimed sublinear advantages.

W2: Symmetrization Limitation: Because the algorithm approximates the attention matrix through a symmetric kernel on the combined $(Q, K)$ dataset, it effectively provides guarantees only for the symmetrized form $(A + A^\top)/2$. This design choice limits its direct interpretability as an approximation to the true attention matrix, and it remains unclear whether the same speedup can be achieved without this symmetrization.

**Questions:**

Q1:Regarding Empirical Validation (W1):
Could the authors provide any empirical or simulated evidence to illustrate the practical implications of the proposed algorithm? For example, could a small-scale classical simulation or synthetic experiment demonstrate the expected sublinear scaling behavior or approximation quality?

Q2:Regarding Symmetrization Limitation (W2):
The current framework provides guarantees only for the symmetrized attention matrix $(A + A^\top)/2$. Do the authors believe the same quantum speedup could be achieved without this symmetrization? If not, could they elaborate on the fundamental technical barriers that make direct approximation of the asymmetric attention matrix more challenging?

---

> ### Author Response · Authors · 2025-11-15
>
> We appreciate your valuable comments, in particular your encouraging acknowledgement that our work is highly original, has strong theoretical significance and takes advantage of several sophisticated algorithmic paradigms to provide a quantum speedup. We have significantly improved the results obtained in the paper and we refer you to the global response for an overview. Below, we address your specific concerns.
>
> * Regarding empirical evaluations: Thank you for raising this question, although our algorithm would require QRAM to empirically verify its quadratic speedup, we perform some preliminary experiments verifying that various assumptions indeed hold. We refer you to the global response for more details.
>
> * Regarding symmetrization limitation: Thank you again for raising this point, in our revision, we have successfully removed the symmetrization guarantee, in short, instead of using the stronger spectral approximation guarantee which would yield the symmetrization error, we show that a weaker bound on spectral and Frobenius norm for $A-\widetilde A$ is sufficient to obtain the desired error guarantee. We refer you to the global response for more details. We have updated the manuscript respectively to reflect the changes (see page 7, line 356 - 368).

---

> > ### Comment · Reviewer_zcdq · 2025-11-17
> > **Regarding experiments**
> >
> > I commend the authors for actually adding experiments, but I also want to "defend them" and to mention that I find the request unreasonable. This is clearly a theoretical work. The goals, the proofs, and the claims are pretty clear. Experiments (or "more experiments") are always nice to have, but let us first evaluate the papers with what they already provide, before asking for additional content (where the reviewers have to spend more time reviewing this new content).

---

> > > ### Comment · Reviewer_ppDN · 2025-11-26
> > >
> > > Thank you for the detailed clarifications and the substantial revisions.
> > > Regarding the experimental aspect: although I did list the limited empirical evaluation as a weakness, my overall assessment was based primarily on the theoretical soundness of the work, and I appreciate that the paper is fundamentally theoretical in nature. The added experiments and the explanation of the underlying assumptions were helpful and properly addressed my concerns.
> > > Thank you again for the thoughtful response — I have updated my score for Soundness accordingly.

---

> > > > ### Author Response · Authors · 2025-11-26
> > > >
> > > > Thank you for the update. We would also like to note that regarding the theory aspect, we have removed the symmetrization requirement by observing that a weaker, spectral norm and Frobenius norm approximation is enough to obtain the final guarantee, see global response for more details. We are happy to address any additional comments or concerns you have. Thank you again.

---

### Author Response · Authors · 2025-11-15
**Global response**

We thank all the reviewers for your helpful and constructive comments and we have significantly improved the results obtained in this work. We summarize them below, and they have been revised accordingly in the manuscript (marked in red).

* Approximation guarantee without symmetrization: in the original draft, we obtain an approximation in terms of the symmetrization of the $A:=\exp(QK^\top)$ matrix in the following form: $\\|\widetilde D^{-1} (\widetilde A+\widetilde A^\top)/2 \widetilde V - D^{-1} (A+A^\top)/2 V\\|_F$, in the revision, we successfully obtain the approximation guarantee in terms of the true attention: $\\|\widetilde D^{-1} \widetilde A \widetilde V - {\rm Att}(Q, K, V)\\|_F$ where ${\rm Att}(Q, K, V)=D^{-1}AV$. Our main observation is that to obtain the desired guarantee, it is enough to bound the spectral and Frobenius norm error $\\|A-\widetilde A\\|$ and $\\|A-\widetilde A\\|_F$ while in the original version, we have to rely on a much stronger spectral approximation guarantee $A+A^\top\preceq \widetilde A+\widetilde A^\top \preceq A+A^\top+2\lambda I$. Our new guarantee is much more natural and lets one directly interpret the quality with respect to the attention matrix.

* Relaxed condition on the matrix $D^{-1}$: in the original draft, in order to obtain a meaningful approximation guarantee, we require $\\|D^{-1}\\|\leq \frac{1}{\epsilon \\|E\\|+\lambda\sqrt{n}}$ where $E$ is the exponential kernel over the dataset $Q\cup K$. In the revision, we improve our analysis and relax the requirement to $\\|D^{-1}\\|\leq \frac{1}{\epsilon \\|A\\|+\lambda \sqrt{n}}$, this requirement is much weaker as $A$ is the off-diagonal block of $E$ and typically has a much smaller spectral norm than $E$. To realize this new bound, we developed a tailored analysis of the quantum matrix-vector product approximation algorithm that replaces the dependence on $\\|E\\|$ by $\\|A\\|$.

* Refined upper bound on the row distortion $\alpha$: in the original draft, we didn’t provide an upper bound on $\alpha$, which appears in our runtime bound. This naturally raises the question whether $\alpha$ can be in the order of $n$ and hence diminishes the quadratic advantage we obtain. Thanks to Reviewer zcdq who points out that $\alpha$ admits an upper bound of $d$, we refine the argument and prove that $\alpha(V)\leq \frac{d}{{\rm srank}(V)}$ where ${\rm srank}(V)=\frac{\\|V\\|_F^2}{\\|V\\|^2}$ is the stable rank of $V$. This means that for the setting we are interested in where the sequence length is much larger than the key-value dimension, $\alpha$ is strictly sublinear.

* Empirical verification on the assumptions of parameters: while our algorithm requires QRAM and thus is hard to empirically verify its efficiency, we instead check several assumptions on the parameters. These include (1) the assumption that $\\|D^{-1}\\|\leq \frac{1}{\epsilon \\|A\\|+\lambda \sqrt{n}}$, we see that this condition is satisfied in most cases for any $\epsilon\leq 0.18$, (2) Frobenius norm vs spectral norm, specifically, we check the Frobenius and spectral norms of $A$ and $V$. For $A$, we empirically show that the Frobenius norm is typically within a factor of 2 of the spectral norm, while for $V$, it does have the $\sqrt{d}$ gap predicted by the theory. We note that to cancel out the $\sqrt{d}$ factor, we can scale down $\epsilon$ by a factor of $\sqrt{d}$, causes a $\sqrt{d}$ runtime blowup, maintaining the sublinear behavior, (3) the upper bound on the row distortion factor $\alpha(V)$, $d/{\rm srank}(V)$, which we observe is a small constant less than 3, (4) infinity norm vs spectral norm for $A$, which we again observe that they are only a constant factor ($<3$) off rather than $\sqrt{n}$. Note that the $\lambda \sqrt{n}$ factor in the approximation guarantee largely comes from converting the spectral norm bound to infinity norm bound. In experiments, we show these two quantities are very close for $A$ and do not exhibit a $\sqrt{n}$ gap. These experiments are performed on the pretrained OLMo2 1B and 7B models and evaluated on the pretraining datasets. These models have sequence length $n=4096$ and $d=128$. We aggregate the statistics over all heads and across all layers, and report their means. We record the results in the following table. For more details, see Appendix E in the revised manuscript.

| Metric                                                        |  OLMo2-1B |  OLMo2-7B |
|:-------------------------------------------------------------|----------:|----------:|
| Largest $\epsilon$ s.t. $\|D^{-1}\|\leq \frac{1}{\epsilon \|A\|}$ |   0.1708  |   0.1685  |
| $\frac{\|A\|_F}{\|A\|}$                                     |   1.3769  |   1.3586  |
| $\frac{\|V\|_F}{\|V\|}$                                     |  11.3137  |  11.3137  |
| $\frac{d}{{\rm srank}(V)}$                                  |   1.7126  |   2.1345  |
| $\frac{\|A\|_{\infty}}{\|A\|}$                              |   2.7439  |   2.7439  |

---

### Author Response · Authors · 2025-12-01
**Summary of the discussion period**

Dear Area Chair,

Per the unprecedented situation about open review information leak, ICLR’s decision to revert scores pre-rebuttal and reassign AC, we would like to document our discussion with reviewers during the discussion period.

*  **Reviewer ppDN**: The reviewer points out two major weaknesses of our work pre-rebuttal: (1) the theoretical guarantee is in terms of a symmetrization of the attention matrix, (2) lack of experiments. For (1), we have completely resolved the issue and obtained an error guarantee in terms of the true attention matrix, as detailed in the global response. For (2), the reviewer is satisfied with our experiments verifying the conditions our theorem are required to meet, and has raised the **Soundness score to 3**, as documented in our discussion thread.

*  **Reviewer ojWR**: While the reviewer is not active during the discussion period, the reviewer **provided an initial positive rating of 8**.

*  **Reviewer eeim**: The reviewer is mainly concerned about the dependence on parameters such as $\alpha$, which could be large in practice, and the assumption on $\\|D^{-1}\\|$. For the former, we prove that $\alpha\leq \frac{d}{{\rm srank}(V)}$, where ${\rm srank}(V)$ is the stable rank of $V$, so it is at most $d$ as ${\rm srank}(V)\geq 1$. For the latter, we conduct numerical experiments to show that for practical models, it is enough to pick any $\epsilon\leq 0.18$ to satisfy the $\\|D^{-1}\\|$ assumption. The reviewer is satisfied by our response and **maintains the positive score 6**.

*  **Reviewer zcdq**: The reviewer has two main concerns: (1) for QRAM model, a comprehensive bit complexity analysis is needed, as one only has $O(\log n)$ qubits in this model, (2) the approximation guarantee “hides” factors that could depend on $n$, such as Frobenius norm vs spectral norm. For the first concern, we have a long thread of detailed discussion on how to prove the bit complexity of our algorithm, and we have shown that at the expense of polylogarithmic factors, the overall runtime of our algorithm is preserved. We appreciate the reviewer for pointing out many useful references for this discussion. For the second concern, we numerically show that the Frobenius norm is only off by a constant factor of the spectral norm. The reviewer is very satisfied with the discussion and **has raised the score from 4 to 8**, as documented in our discussion thread.

We hope this further clarifies the reviewers’ stances regarding our rebuttal, thank you!

---

### Meta-Review · Area_Chair_PKbQ · 2025-12-28

**Summary:**

This paper proposed a quantum algorithm for approximating the attention matrix. Calculating the attention matrix is fundamental in current LLMs based on transformers, and this paper proposed the first quantum algorithm that achieves sublinear time complexity.

In the reviews, the reviewers are impressed by the significant contributions in theory, both its importance and rigorousness. However, there are also some concerns raised:
- The initial version provided guarantees only for the symmetrized form $(A+A^{\top})/2$.
- Parameter dependences, which influences the applicability of the result as well as the tradeoff between approximation/complexity.
- The paper is purely theoretical, without experimental demonstrations.

During the rebuttal, the authors significantly revised the paper and the new version included:
- Approximation guarantee without symmetrization.
- Relaxed condition on the matrix.
- Refined upper bound on the row distortion.
- Empirical verification on the assumptions of parameters.

The reviewers found these revisions very helpful. Although we encountered the difficult situation this year, it is reached into consensus that this paper should be accepted as:
- Reviewer ppDN was satisfied with the new experiment results.
- Reviewer ojWR: The reviewer kept an initial positive rating of 8.
- Reviewer eeim: The reviewer replied the authors and kept the positive score 6.
- Reviewer zcdq: After very adequate discussions, the reviewerraised the score from 4 to 8.

The meta-review hence follows the accept decision. Note that this paper was significantly revised during the rebuttal, and the authors should merge all the points raised during the rebuttal into the final version.

**Reviewer Concerns:**

The authors did an excellent job on addressing reviewer concerns, and I don't see remained outstanding issues. See the meta-review summary.

**Reviewer Scores:**

This question does not apply to this paper - fortunately, there was adequate discussions between the authors and the reviewers. In particular:
- Reviewer ppDN was satisfied with the new experiment results.
- Reviewer ojWR: The reviewer kept an initial positive rating of 8.
- Reviewer eeim: The reviewer replied the authors and kept the positive score 6.
- Reviewer zcdq: After very adequate discussions, the reviewerraised the score from 4 to 8.

---

### Decision · Program_Chairs · 2026-01-26

Accept (Poster)